# AccLoRT: Pretraining and tuning of Large Language Models from accumulation of low-rank weights

## Abstract

Pretraining large language models (LLMs) poses significant computational challenges, particularly due to the memory requirements that exceed the capabilities of standard GPU devices. To address these issues, we introduce a fully low-rank approach for LLMs pretraining to improve the memory efficiency. Specifically, our approach sequentially trains low-rank matrices and accumulates them into a frozen high-rank matrix until convergence. Notably, our approach enables the low-rank traning without a warm up phase with full parameter, therefore achieving memory efficiency in the entire training process. We provide a comprehensive theoretical analysis for our proposed method by establishing the upper and lower bounds for the rank of multiple matrix sums and analyzing the rank dynamics in low-rank adapters. The results show that with finite accumulation steps, the accumulated low-rank training is equivalent to full-rank training. Extensive experiments on both synthetic reduced rank regression and practical Llama models (60M to 1B parameters) validate the effectiveness of the proposed approach in pretraining, demonstrating its potential to make LLM development more accessible and efficient.

## 1 Introduction

Recent advances in large language models (LLMs) have demonstrated that scaling up model parameters leads to remarkable improvements in model performance, as evidenced by state-of-the-art models like GPT-4 (Achiam et al., 2023), PaLM 2 (Anil et al., 2023), Claude, and Llama 3 (Dubey et al., 2024), which contain hundreds of billions of parameters. However, pretraining such large-scale models has become increasingly challenging due to the enormous computational demands, particularly the memory requirements that far exceed the capabilities of standard GPU devices. For instance, training a 7B Llama architecture with bfloat16 parameter and single batch size requires approximately 60GB of GPU memory for weight parameters, Adam optimizer states and weight gradients, which is beyond the capacity of many single GPU devices (Zhao et al., 2024). This computational barrier has made it practically infeasible for researchers with limited resources to participate in fundamental LLM pretraining.

Despite the resource constraints and memory bottlenecks highlighted during pre-training, researchers have started exploring ways to reduce the number of post-training parameters for downstream tasks, thereby alleviating the computational burden on limited hardware (e.g., a single GPU). Following this line of thought, parameter-efficient fine-tuning (PEFT) (Houlsby et al., 2019) approaches have been proposed to diminish memory and computational demands without significantly compromising model performance. Among these methods, Low-Rank Adaptation (LoRA) (Hu et al., 2021) has garnered particular attention by inserting learnable low-rank matrices into a pre-trained large model (PLM) and updating only a small subset of parameters. In many well pretrained PLM, LoRA achieves near-full fine-tuning performance on various downstream tasks.

While low-rank parameterization offer significant memory efficiency benefits, adopting such approaches for LLM pre-training presents substantial challenges, as demonstrated by observed performance degradation (Wei et al., 2024). This limitation aligns with empirical expectations, as weight matrices in pre-trained language models typically exhibit high-rank or full-rank weight characteristics.

Several very recent studies have attempted to address this challenge by using low-rank approximation on gradient (Zhao et al., 2024) or weight matrices (Ren et al., 2024; Han et al., 2024). Despite these advances, current approaches face significant limitations in achieving true memory and parameter efficiency. Methods like GaLore (Zhao et al., 2024) still require maintaining full weight parameters and computing complete gradients throughout training, while approaches such as ReLoRA (Lialin et al., 2024) necessitate full-parameter warmup pre-training before transitioning to low-rank training in the final

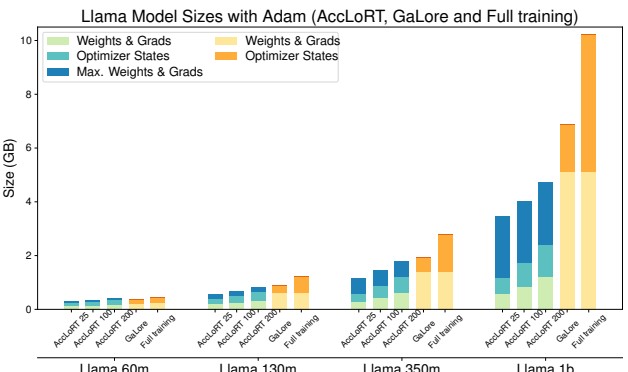

Figure 1: Sizes of different Llama architecture with Adam optimization for AccLoRT, GaLore and full pretraining.

phase, effectively incurring computational costs equivalent to traditional full pre-training. Motivated by these limitations, a fundamental question emerges: *is it possible to achieve full-rank training while achieving memory efficiency in the entire training process?*

In this paper, we introduce an AccLoRT approach to pretrain LLMs with fully low-rank training to maintain memory efficiency. The key idea exploits the mathematical property that, when trained parameters span multiple distinct subspaces, the summation of their corresponding low-rank matrices yields a matrix of higher rank. Therefore, we can train sequentially low-rank matrices and accumulate them in a frozen matrix of increasing rank until reaching convergence. To establish a theoretical foundation for AccLoRT, we first derive fundamental upper and lower bounds for the rank of matrix sums. Building upon these results, we characterize the rank evolution dynamics in low-rank adapters training and demonstrate that for independent random matrices, their summation achieves a rank equal to the minimum between their cumulative ranks and the maximal achievable rank. The main contributions can be summarized as

- We propose AccLoRT, a new approach that enables memory-efficient pre-training of LLMs with fully low-rank training. The approach progressively accumulates and freezes learned parameters while introducing new low-rank components, thereby maintaining consistent memory efficiency while achieving effective model training through exclusively low-rank operations.

- Our theoretical investigation into AccLoRT present the rank dynamics in low-rank adapters and precise characterization of rank evolution across AccLoRT layer components, ultimately establishing equivalence to full-rank training after finite accumulation steps.

- We conduct extensive empirical evaluations through both synthetic experiments with reduced rank regression and comprehensive pre-training experiments on Llama models across multiple scales (60M to 1B parameters), demonstrating the approach's effectiveness and scalability in both synthetic and practical settings.

## 2 RELATED WORKS

**Low-rank adapters and compression** Low-Rank Adaptation (LoRA) (Hu et al., 2021) has emerged as a breakthrough approach for efficient large language model fine-tuning. The method works by freezing pretrained weights while introducing trainable low-rank adapter matrices for each linear layer, substantially reducing the memory footprint during fine-tuning tasks. Since its introduction, the field has seen significant advancements across multiple directions: hyperparameter optimization (Hayou et al., 2024; 2025), enhanced training strategies (Liu et al., 2024b; Yang et al., 2024a; Jin et al., 2024), novel regularization techniques (Meng et al., 2024; Wang et al., 2024; Ji et al., 2024), sparsity implementations (Ding et al., 2023; Bhardwaj et al., 2024; Panda et al., 2024), weight decomposition (Liu et al., 2024a; Kopiczko et al., 2024), and union of subspace Hameed et al. (2024); Lialin et al. (2024). In addition to the low-rank adapters, recent works have also demonstrated the great success of low-rank compression for LLMs, which aims to approximate the pretrained weights to reduce

inference latency and storage usage. A major challenge in decomposing LLMs lies in the massive outliers present in activations. To address this, Yuan et al. (2023) proposed ASVD, which scales the weight matrices based on activation channels before applying SVD. Building on this, SVD-LLM (Wang & et al., 2024) incorporates a truncation-aware low-rank estimation algorithm to further preserve model capabilities. More recently, researchers have investigated the orthogonality between low-rank decomposition and other compression techniques, such as SliceGPT (Ashkboos et al., 2024), which achieves structured pruning through computational invariance, effectively reducing the rank of weight matrices in a structured manner.

**Efficient Pretraining for LLMs**  Different from PEFT, pretraining for LLMs is more challenging due to the high memory consumption of the optimizer states and gradients. To address this challenge, several recent works have proposed methods to reduce memory consumption during pretraining. The first line of these approaches is to optimize the weight matrices in a low-rank subspace (Lialin et al., 2024; Han et al., 2024; Mo et al., 2025). These methods typically employ a two-stage training strategy with full parameter warm-up, or simultaneously train low-rank plus sparse matrices, which increases the memory cost or training complexity. The second line is to project the gradient into a low-rank subspace to reduce the optimizer state memory in Adam (Zhao et al., 2024; zmu, 2025; Chen et al., 2025; Zhu et al., 2024). However, the low-rank projection for the gradient may incur the inevitable suboptimal performance. To address this challenge, Chen et al. (2025) introduced both low-rank projection and complement subspace projection for gradient, enabling promising high-rank update for each iteration, and achieved superior performance. To further reduce the peak memory value on full SVD for projection matrices, Zhu et al. (2024) introduced the random projection for calculating the scaling factor, and rescaled the gradient to perform an SGD-like update.  Compared with these methods, our proposed AccLoRT approach trains low-rank matrices from scratch and accumulates them into a frozen matrix with increasing rank, achieving high-rank training without requiring full weight storage or computation.

## 3 THEORETICAL BACKGROUND

### 3.1 BACKGROUND & LOW-RANK ADAPTERS

LoRA emerged from the simple idea that the fine-tuning process possesses a low intrinsic rank (Aghajanyan et al., 2020). To take advantage of this statement, they suggested updating a pre-trained weight matrix $W \in \mathbb{R}^{n \times m}$ with low-rank changes during fine-tuning. This is modeled by freezing the weight matrix and using two low-rank matrices $B \in \mathbb{R}^{n \times r}, A \in \mathbb{R}^{r \times m}$ to represent the layer as

$$y = (W + BA)x,$$

where only $A$ and $B$ are trainable. This results in very low memory needed for the finetuning as only the small matrices $A$ and $B$ will account for the gradients and the optimizer states.

### 3.2 BOUNDS FOR THE RANK OF THE SUM OF MATRICES

While the fundamental bounds for the rank of a sum of two arbitrary matrices were established in the seminal work (Marsaglla, 1964), the characterization of rank behavior under multiple matrix additions remains poorly documented. We address this gap by generalizing this result to finite matrix sums and derive the upper and lower bounds for the rank of the sum of a finite family of matrices. These results will be instrumental in our analysis of the rank behavior of the proposed AccLoRT.

**Proposition 3.1** (Upper bound). *Let $(A_i)_{1 \leq i \leq d}$ be a finite family of matrices in $\mathbb{R}^{m \times n}$. Then, we have the upper bound for the rank of their sum:*

$$\text{rank}(\sum_{i=1}^{d} A_i) \leq \min(\text{rank}([A_1 \ \cdots \ A_d]), \text{rank}([A_1^\top \cdots A_d^\top]^\top)).$$

**Proposition 3.2** (Lower bound). *Let $(A_i)_{1 \leq i \leq d}$ be a finite family of matrices in $\mathbb{R}^{m \times n}$. Then, we have the lower bound for the rank of their sum:*

$$\text{rank}(\sum_{i=1}^{d} A_i) \geq \text{rank}([A_1 \ \cdots \ A_d]) + \text{rank}([A_1^\top \cdots A_d^\top]^\top) - \sum_{i=1}^{d} \text{rank}(A_i).$$

Complete proofs are to be found in Appendix E.

## 3.3 RANK OF LORA UNDER STOCHASTIC GRADIENT DESCENT

To our knowledge, the theoretical analysis of rank evolution in LoRA during training remains unexplored in the literature. While heuristic arguments suggest that zero-initialized matrices should evolve toward full rank under gradient descent optimization, while randomly initialized matrices maintain their initial full-rank property, rigorous theoretical results characterizing these rank dynamics are currently lacking. This gap in understanding motivates our investigation into the mathematical principles governing rank evolution in low-rank neural network training.

**Theorem 3.3.** *Consider any neural network under stochastic gradient descent optimization with a LoRA layer $y = Wx + ABx$ with $W \in \mathbb{R}^{n \times m}$ frozen and a rank $r < \min(n, m)$. Suppose that the training dataset has no duplicates and that the initial matrices are $A_0 = 0$ and $B_0$ such that its elements are drawn from a distribution absolutely continuous with respect to the Lebesgue measure. Then, with probability one, for a batchsize $d$, at time $t$,*

$$\text{rank}(A_t) = \min(d \times t, r) \quad \text{rank}(B_t) = r$$

The proof can be consulted in Appendix H. An illustration of the theorem is provided in Figure 4. This theorem characterizes the distinct rank evolution patterns of matrices $A$ and $B$ in LoRA layers during training. Specifically, while matrix $B$ maintains its initial full rank $r$ throughout the optimization process, the rank of matrix $A$ grows linearly with training steps at a rate determined by the batch size, until reaching the predefined rank $r$. This result formally establishes that LoRA's rank dynamics follow a predictable trajectory under stochastic gradient descent, providing theoretical justification for the empirically observed behavior of low-rank adapters.

## 4 ACCLORT METHOD

In this section, we propose AccLoRT, a novel training methodology that progressively accumulates low-rank adaptations into frozen high-rank weight matrices through periodic merging operations, followed by adapter reinitialization. This approach enables efficient approximation of full-rank equivalent weights during the pretraining phase while maintaining efficiency for both parameter and memory compared to (Zhao et al., 2024; Lialin et al., 2024).

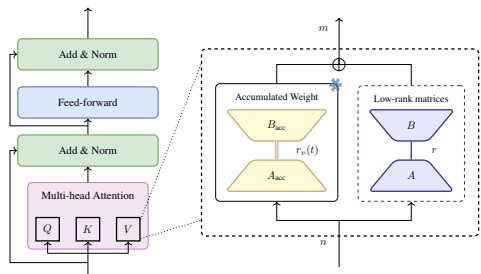

Figure 2: Accumulated Low-rank Training (AccLoRT) layer

### 4.1 ACCUMULATION OF LOW-RANK MATRICES

Random matrices have important properties that help understand the idea of accumulating low-rank matrices to reach full-rank. The following proposition shows that the rank of a sum of low-rank matrices will be equal to the sum of all the ranks bounded by the maximal reachable rank $min(n, m)$ for matrices of shape $n \times m$.

**Proposition 4.1.** *Let $n, m \in \mathbb{N}^*$ and $r \leq \min(n, m)$ be an integer, $(A_1, B_1), \ldots, (A_d, B_d)$ be $d$ pairs of i.i.d random matrices in $\mathbb{R}^{n \times r} \times \mathbb{R}^{r \times m}$. Then, with probability 1, each matrix $A_i B_i$ has rank $r$ and*

$$\text{rank}\left(\sum_{i=1}^{d} A_i B_i\right) = \min(n, m, rd)$$

.

The proof can be found in Appendix F.

It may not be totally clear that the weight matrices are still random enough after some optimization steps to fulfill this property. However, each matrix $A$ or $B$ would be updated as $A := A - \eta \cdot \text{Opti}(\nabla_A \mathcal{L}(W), \cdots)$ where the $\cdots$ accounts for optimizer states like first and second moments in Adam (Kingma & Ba, 2017). As in the case of training LLMs, we usually use large datasets such as the C4 dataset (Raffel et al., 2023) where samples are never seen again, thus guaranteeing null intersection of the row and column space between the matrix and the update term. A more detailed discussion will be made in the case of (Stochastic) Gradient Descent in section 4.3.

---

**Algorithm 1** Detailed of Accumulated Low-rank Training (AccLoRT) with Adam

---

**Input:** The rank $r$, weight matrices $W_{\text{acc}} \in \mathbb{R}^{n \times m}$, boolean reset_scheduler.
Initialize the random matrices $A \in \mathbb{R}^{n \times r}$ and $B \in \mathbb{R}^{r \times m}$ using QR decomposition on a random $W_0$, the Adam optimizer first-order moments $m_A \in \mathbb{R}^{n \times r}$, $m_B \in \mathbb{R}^{r \times m}$ and second-order moments $v_A \in \mathbb{R}^{n \times r}$, $v_B \in \mathbb{R}^{r \times m}$ to zero matrices, and set the optimizer step counters $t_A, t_B$ to zero.
**while** not converged **do**
   **if** t % T = 0 **then**
      $r_{\text{virtual}} \leftarrow r \lfloor t/T \rfloor$, $W_{\text{acc}} \leftarrow A_{\text{acc}} B_{\text{acc}} + AB$       $\triangleright$ Accumulated low-rank approximation
      **if** $r_{\text{virtual}} < \min(n, m)$ **then**
         $A_{\text{acc}}, B_{\text{acc}} \leftarrow \text{QR}_{r_{\text{virtual}}}(W_{\text{acc}})$       $\triangleright$ Truncated QR decomposition
      **else**
         $A_{\text{acc}}, B_{\text{acc}} \leftarrow W_{\text{acc}}, I_m$
      **end if**
      Draw $B$ i.i.d. random matrix in $\mathbb{R}^{r \times m}$ and orthogonalize it using QR decomposition, and set $A \leftarrow 0$
      **if** reset_scheduler **then**
         $m_A, v_A \leftarrow 0; m_B, v_B \leftarrow 0; t_A, t_B \leftarrow 0$
      **end if**
   **end if**
   Update $A$ and $B$ using Adam optimizer, and let $t \leftarrow t + 1$
**end while**
**Output:** $W_{\text{acc}}$

---

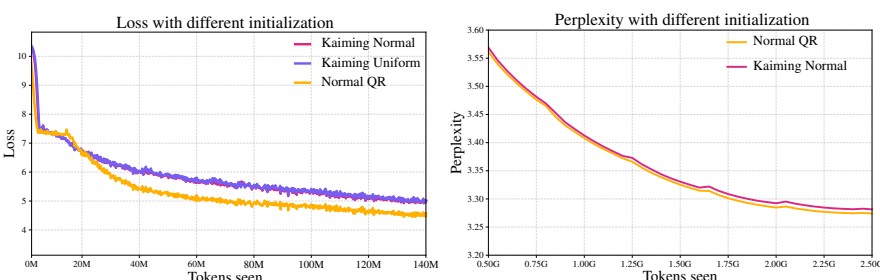

Figure 3: Loss and perplexity for different initialization methods on Llama 60M and 130M.

## 4.2 METHOD

**AccLoRT layer** We define a AccLoRT linear layer as a low-rank adapter with $W_{\text{acc}}$ a frozen weight and two trainable matrices $A \in \mathbb{R}^{n \times r}, B \in \mathbb{R}^{r \times m}$ parametrized by a rank $r$ as well as an accumulation frequency $T$. The main difference with LoRA lies in the dynamicity of the AccLoRT layer. After each $T$ training steps, the low rank matrices are frozen and merged into $W_{\text{acc}}$, then, new low-rank matrices will be drawn. This can be summarized at step $t$ by the equations

$$y = W_{\text{acc}} x + ABx, \text{ with frozen } W_{\text{acc}} = \sum_{i=0}^{\lfloor t/T \rfloor} A_i B_i,$$

where $A_i, B_i$ are the trainable low-rank matrices drawn after the $i + 1$-th accumulation. Note that at the beginning of the pretraining, the layer is simply $y = ABx$ as $W_{\text{acc}} = 0$. Figure 2 illustrates the AccLoRT layer for the weight matrices of a multi-head attention layer where each linear sublayers are replaced with a AccLoRT layer.

**Initializing low-rank matrices** In AccLoRT, initialization occurs at two distinct stages: the initial step ($t = 0$) and subsequent accumulation steps ($\lfloor t/T \rfloor > 0$). For subsequent accumulation steps, we follow the initialization strategy of LoRA (Hu et al., 2021), setting adapter matrix $A$ to zero while randomly initializing $B$ to maintain training continuity. However, the initialization at $t = 0$ requires careful consideration.

Classical initialization techniques such as the Xavier Glorot (Glorot & Bengio, 2010) or the Kaiming He (He et al., 2015) initialization have shown to have important repercussions on the training and performance of neural netowrks. Llama architecture (Touvron et al., 2023a) used a much simpler

initialization which is to set all linear weights of the feed forward networks and matrices of attention mechanism to be normaly distributed with standard deviation of 0.02. *One question that arises is how shoud we initialize our weights in the case of AccLoRT at the beginning run?*

Indeed, in contrast to finetuning tasks, AccLoRT layers need to have both adapter matrices $A, B$ to be randomly initialized to ensure that the layer output is not zero as $W_{\text{acc}} = 0$ at the beginning of the training. However, choosing a right distribution for both of them is not as straightforward as it seems if we wish to reach efficiency. There is no known result for the distribution of the product of two independent random matrices with normal i.i.d entries (most results are asymptotical ones (Götze et al., 2017; Kopel et al., 2020; Shen, 2001; Akemann et al., 2013; Mehta, 2004))

Fortunately, it is easy enough to draw matrices $A$ and $B$ in such a way that their product is close enough to the normal initialization. We find it more suitable to our method to perform a truncated QR decomposition on an actual normal matrix with the same parameter as what should have been chosen for a full-rank weight matrix and set both matrices $A$ and $B$ to be the QR matrices. The matrix $A$ will then follow a Haar distribution if the QR decomposition has been implemented using the Graam-Schmidt algorithm (Mezzadri, 2007) while the $B$ matrix will have its diagonal element chi-square distributed. This initialization approach is very effective, and results in lower perplexity of the trained models compared to other initialization as described in Figure 3 for both Llama 60m and 130m models.

Table 1: Parameter breakdown for different methods (AccLoRT with $d$ accumulations, GaLore, LoRA, and full-parameter pretraining). The total footprint is decomposed into frozen parameters, trainable parameters, optimizer states, and gradients.

| Method | AccLoRT | GaLore | LoRA | Full |
|---|---|---|---|---|
| Frozen parameters | $\min(nm, (n+m)rd)$ | - | $nm$ | - |
| Trainable parameters | $(n+m)r$ | $nm$ | $mr + nr$ | $nm$ |
| Optimizer states | $2(nr + mr)$ | $ms + 2ns$ | $2mr + 2nr$ | $2nm$ |
| Gradients | $(nr + mr)$ | $nm$ | $mr + nr$ | $nm$ |
| Total | $\min(nm, (n+m)rd) + 3r(n+m)$ | $2nm + ms + 2ns$ | $nm + 3r(n+m)$ | $4nm$ |

**Weight Accumulation & Memory Optimization**   From Section 4.1, achieving full-rank representation in $W_{\text{acc}}$ requires at least $\lceil \min(n, m)/r \rceil$ accumulations, as each accumulation increases the rank by $r$ from an initial rank of zero. To optimize memory usage, we can express $W_{\text{acc}}$ as the product of two low/medium-rank matrices, $A_{\text{acc}} B_{\text{acc}}$. To this end, we propose two methods. The first one is represented in Algorithm 1: when full-rankness is not achieved for the given AccLoRT layer, to preserve the model from recovering a full size matrix $W_{\text{acc}}$, we split it in two parts $A_{\text{acc}}, B_{\text{acc}}$ as the $r\lfloor t/T \rfloor$-truncated QR decomposition of $W_{\text{acc}}$. A second and more efficient way is simply to set $A_{\text{acc}}, B_{\text{acc}}$ to be the block matrices defined by the previous frowen low-rank matrices:

$$A_{\text{acc}} = [A_0 \mid \cdots \mid A_p], \quad B_{\text{acc}} = [B_0^\top \mid \cdots \mid B_p^\top]^\top$$

after the $p$-th accumulation. This structured decomposition significantly reduces memory overhead, making it a more scalable alternative to the full weight matrix.

**Number of Parameters**   The parameter efficiency of AccLoRT closely aligns with that of LoRA, as only low-rank adapters are actively trained at any given time. While this fundamental efficiency is preserved through the accumulation process, AccLoRT introduces a key distinction during pre-training. Unlike fine-tuning, where we utilize either the full or QR-truncated pretrained weights, pre-training begins with an empty accumulation matrix whose rank progressively increases through weight optimization. Consequently, the memory footprint of the non-trainable weights in an AccLoRT layer is bounded by $\min(mn, (n+m)rd)$, where $d$ represents the number of accumulated updates. Table 1 illustrates this and compares the number of parameters for different methods such as GaLore and LoRA.

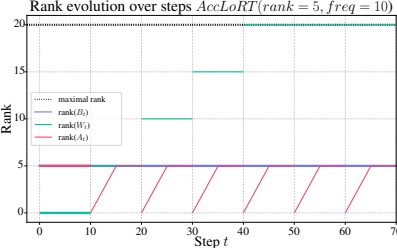

Figure 4: Rank evolution of matrices $A, B, W_{\text{acc}}$ for frequency of $T = 10$ steps, rank 5 and batchsize 1. The maximal rank is 20.

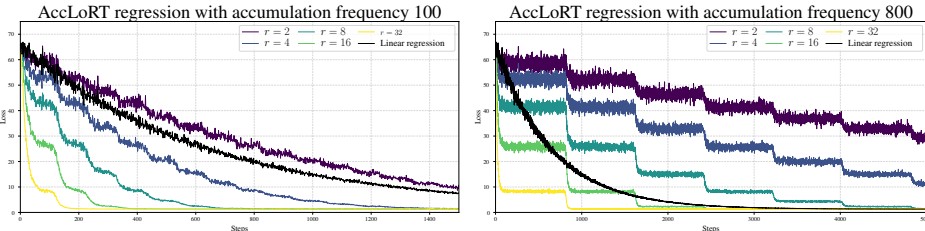

Figure 5: Loss of AccLoRT linear regression for different accumulation frequencies and ranks vs standard linear regression with SGD optimization.

**ReLoRA vs AccLoRT**   While our approach seems to closely match ReLoRA (Lialin et al. (2024)), there are some fundamental differences that makes the two methods mathematically and algorithmically different. First, during the optimization, while ReLoRA performs a full-rank warmup followed by Kaiming/zero-initialized adapters, AccLoRT samples a full-rank Gaussian matrix at $t = 0$, takes its truncated-QR decomposition on the first $r$ subspaces of singular values, and sets $B = Q_r, A = R_r$. After every accumulation it repeats this QR-based sampling to define a fresh orthonormal subspace, which yields more stable low-rank training and faster loss/perplexity reduction. AccLoRT and ReLoRA also learn in different regimes: ReLoRA behaves like a finetuning after the full-rank warmup, whereas AccLoRT explicitly fills the parameter space through successive orthogonal low-rank subspaces before reaching full rank. Finally, AccLoRT is practically simpler and more stable: it avoids full-rank warmup, jagged cosine restarts, pruning-based optimizer resets, and delicate tuning, relying only on the choice of rank and accumulation frequency.

### 4.3 ASYMPTOTICAL HIGH-RANK EQUIVALENCE OF ACCLORT

The rank evolution patterns established in Theorem 3.3 for LoRA training dynamics, combined with the rank additivity properties demonstrated in Proposition 4.1, provide a theoretical framework for analyzing the rank behavior in AccLoRT layers. This leads to the following corollary, which characterizes the precise rank evolution across all components of a AccLoRT layer.

**Corollary 4.2.** *For a AccLoRT layer at step $t$ with accumulation frequency $T$ and rank $r$,*

$$\text{rank}(A_t) = r\mathbb{1}_{t<T} + \min(dt \mod T, r)\mathbb{1}_{t \geq T}; \quad \text{rank}(B_t) = r; \quad \text{rank}(W_t) = \left\lfloor \frac{t}{T} \right\rfloor r.$$

Figure 4 shows this result for a AccLoRT layer parametrized by a rank of 5 and an accumulation frequency of 10 steps and a batchsize of 1. While the matrix $B$ stays full-rank, the matrix $A$ is initialized full-rank and after the first accumulation reset to zero and increases its rank by $d = 1$ after each step until reaching full-rank and being resetted again after the next accumulation. Concurrently, the matrix $W_{\text{acc}}$ gains $r$ degree of freedom after each accumulation until it reaches its full-rank. From this moment, the new accumulations will update the previously independently optimised dimensions.

## 5 EXPERIMENTS

### 5.1 ACCLORT LINEAR REGRESSION

The Eckart-Young-Mirsky theorem (Eckart & Young, 1936; Mirsky, 1960) states that the best rank-$r$ approximation to a matrix for both the spectral and Frobenius norm is the matrix formed by keeping only the top $r$ left and right singular vectors of the SVD decomposition of the matrix we aim to optimize. The situation becomes slightly more complex when dealing with the best low-rank solution of a linear regression problem as the solution depends on the input dataset. Nevertheless, this problem commonly known as the reduced rank regression (Izenman, 1975) has been thoroughly studied. The proposed AccLoRT can be seen as a generalization of the reduced rank regression problem with accumulated weights:

$$\min_{A_N, B_N} \|Y - W_N X - A_N B_N X\|_F, \quad \text{s.t} \quad W_N = W_{N-1} + A_{N-1}^* B_{N-1}^*, W_0 = 0,$$

where the problem is formulated recursively with $A_{N-1}^* B_{N-1}^*$ the optimal solution of the previous problem (for the sake of clarity and brevity, we consider them to be unique). This modelises the AccLoRT optimization for the mean square problem. For $N > \lfloor \min(n,m)/r \rfloor$, a full-rank solution is learnt and we reach a minima equal to $\|Y - W_{\mathrm{ols}}X\|_2$ as stated by the following theorem.

**Theorem 5.1.** *After $N \geq 0$ accumulation of the AccLoRT linear regression with rank $r$,*

$$\min_{A_N, B_N} \|Y - W_N X - A_N B_N X\|_F \leq \sum_{i > (N+1)r} \sigma_i^2 + \varepsilon_{\mathrm{OLS}}$$

*where the $\sigma_i$'s correspond to the ordered singular values of $W_{\mathrm{OLS}}X = YX^\top (XX^\top)^{-1}X$ and $\varepsilon_{\mathrm{OLS}} = \|Y - W_{\mathrm{OLS}}X\|_2$ is the non reachable error caused by the plausible non-linearity relation between $X$ and $Y$. Additionally, $W_{N+1} = U_{(N+1)r}^\top U_{(N+1)r} W_{\mathrm{OLS}}$ with $U_k$ the first $k$ left singular vectors of $W_{\mathrm{OLS}}X$.*

What this theorem (whose proof can be found in Appendix G) suggests is that after sufficient accumulation the AccLoRT linear regression solution is equivalent to an Ordinary Least Square (OLS) solution. In Figure 5, we can see that the loss of the AccLoRT linear regression decreases faster than the standard linear regression for the same number of iterations for some cases. This is because, at each stage, AccLoRT focuses on optimizing a reduced set of parameters within the current low-rank constraint, allowing for more efficient exploration of the parameter space compared to simultaneous optimization across all dimensions in standard linear regression.

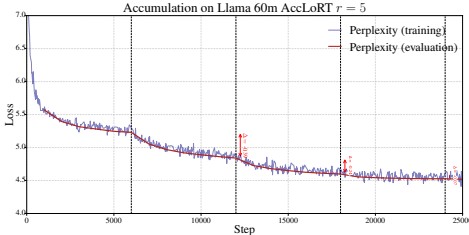

Figure 6: Accumulation dynamic for Llama 60m with 6000 accumulation steps on $r = 5$.

across all dimensions in standard linear regression. More results can be found in the Figure 13 in the Appendix.

## 5.2 ACCUMULATION DYNAMIC ON LLAMA

While the transformer architecture (Vaswani et al., 2023) is utterly more complex than the simple linear regression model that we explored in the previous subsection, we recover similar property experimentally. Intuitively, it is expected that after enough time on a sub-optimization (aka. an optimization before a new AccLoRT accumulation) the model will reach a minimum which is hopefully as global as possible depending on the other hyperparameters. This convergence manifests as a characteristic plateau in the loss curve. Following each accumulation step, the model exhibits renewed convergence with a similar optimization profile as it explores and optimizes over a fresh subspace.

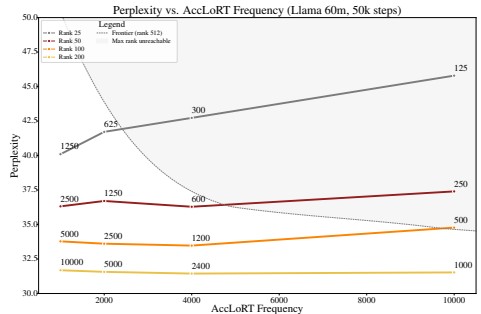

Figure 7: Full-rank frontier and Perplexity vs AccLoRT Frequency for a Llama 60m architecture trained on 50k steps

To demonstrate experimentally our assumption, in Figure 6, we used a small Llama model of 60 million parameters and train a AccLoRT version of it with $r = 5$ to ensure a quick stagnation of the loss with an accumulation frequency of 3000 steps. We train a Llama 60m (see Table 9 for details on the architecture) with a batch size of 256 for 25000 steps. As predicted, the loss starts to plateau after a while as we reached a minimum that fullfill the low rank requirement. After the accumulation, we allow the model to explore 5 more dimensionalities relaunching the learning process and diminishing the loss further.

## 5.3 FULL-RANK PARETO FRONTIER ON PERPLEXITY VS FREQUENCY

To investigate the relationship between the full-rank Pareto frontier, perplexity, and accumulation frequency, we evaluate models under different rank configurations and accumulation frequencies in Figure 7. The numbers annotated in the figure indicate the maximum number of rank-1 updates that

can be stacked before reaching the full-rank boundary. For instance, with rank 25, accumulation frequencies greater than 2441 steps (smaller steps denote higher frequency) prevent the AccLoRT layer from achieving full rank. This limitation is reflected in the performance curves, where crossing the frontier leads to increased perplexity as the model fails to learn representations in unreached subspaces. The results also demonstrate that higher ranks (100 or 200) provide more stable performance across different accumulation frequencies, suggesting better robustness to hyperparameter choices.

## 5.4 PRETRAINING OF LARGE LANGUAGE MODELS

**Baselines, Dataset and Metrics**   We evaluate AccLoRT through extensive pre-training experiments on Llama architectures (Touvron et al., 2023a) ranging from 60M to 1B parameters using the widely-adopted C4 dataset (Raffel et al., 2020). The effectiveness is benchmarked against several state-of-the-art pre-training methods, including full-rank training, GaLore (Zhao et al., 2024), ReLoRA (Lialin et al., 2024), SLTrain (Han et al., 2024), and vanilla LoRA (Hu et al., 2021), with perplexity serving as the primary evaluation metric Jelinek (1977).

Table 2: Model size for weights+gradients and optimizer states in GB. The memory usage of AccLoRT increases with accumulation. We report both the lower and upper bounds.

| Llama | Modules | 60M | 130M | 350M | 1B |
|---|---|---|---|---|---|
| Full-rank | weights + gradients | 0.23G | 0.54G | 1.47G | 5.36G |
| | optimizer | 0.23G | 0.54G | 1.47G | 5.36G |
| GaLore | weights + gradients | 0.23G | 0.54G | 1.47G | 5.36G |
| | optimizer | 0.16G | 0.38G | 0.74G | 2.44G |
| | $r/d$ | 128 / 512 | 256 / 768 | 256 / 1024 | 512 / 2048 |
| AccLoRT | weights + gradients | $0.13 \sim 0.18$G | $0.30 \sim 0.42$G | $0.60 \sim 1.02$G | $1.43 \sim 3.40$G |
| | optimizer | 0.13G | 0.30G | 0.60G | 1.43G |
| | $r/d$/frequency | 100 / 512 / 2000 | 200 / 768 / 5000 | 200 / 1024 / 5000 | 300 / 2048 / 5000 |

**Architecture and training hyperparameters**   In all of our experiments, we used Llama model architecture (Touvron et al., 2023a;b) from the HuggingFace library (Wolf et al., 2020). The main parameters such as hidden size, number of parameters and the number of heads of each architecture are listed in Table 9. The data amount represents the theoretical number of tokens that needs to be fed to the transformer to achieve reasonable convergence. This results is known as the Chinchilla scaling law of large language model (Hoffmann et al., 2022) and states that one needs about 17 times the number of parameters in terms of tokens.

**Performance & Comparison**   We conducted extensive experiments on Llama architectures across multiple scales, from 60M to 1B parameters, comparing our AccLoRT approach with SOTA methods in Table 3. Notably, AccLoRT achieves superior perplexity scores compared to existing approaches, including both memory-efficient methods and full-rank training. For smaller models (60M-350M parameters), AccLoRT consistently outperforms all baselines, reducing perplexity by more than 1.0 compared to GaLore. In the 1B parameter setting, while GaLore achieves the best perplexity of 15.64, AccLoRT maintains competitive performance at 15.49. These results demonstrate that AccLoRT

Table 3: Perplexity of Llama models for different training methods. The baseline results are from Yang et al. (2024b) and Han et al. (2024). Memory usage for baselines and AccLoRT is given in Table 2.

| Llama | 60M | 130M | 350M | 1B |
|---|---|---|---|---|
| AccLoRT | **33.77** | **25.11** | **18.42** | **15.49** |
| SLTrain | 34.15 | 26.04 | 19.42 | 16.14 |
| GaLore | 34.88 | 25.36 | 18.95 | 15.64 |
| ReLoRA | 37.04 | 29.37 | 29.08 | 18.33 |
| LoRA | 34.99 | 33.92 | 25.58 | 19.21 |
| Full-rank | 32.91 | 24.85 | 18.80 | 15.56 |

not only maintains memory efficiency but also achieves SOTA performance, particularly effective for small to medium-sized models where resource constraints are often most critical.

## 5.5 FINETUNING OF LARGE LANGUAGE MODELS

To assess the quality of AccLoRT for finetuning on top of pretraining, we used a widely used GLUE dataset (Wang et al., 2019) comprising multiple tasks on natural language understanding (NLI) on

which we tuned a RoBERTa-base model. Table 4 showcases the effectiveness of AccLoRT for fine-tuning on the GLUE benchmark. Unlike in pretraining scenarios, we observe that AccLoRT does not demonstrate the same overwhelming advantage over LoRA as seen in our pretraining experiments, with performance differences being more modest. This phenomenon likely stems from the inherently low-rank nature of fine-tuning tasks, where the parameter space adaptations required are less complex than in pretraining. Nevertheless, AccLoRT still maintains the highest average accuracy across all tasks, confirming its effectiveness as a PEFT approach that balances computational efficiency with strong task performance. Details in hyperparameter settings can be found in Table 12.

Table 4: Fine-tuning of RoBERTa-base models using AccLoRT on the GLUE dataset.

|  | CoLA | MNLI | MRPC | QNLI | QQP | RTE | SST2 | SSTS-B | Average |
|---|---|---|---|---|---|---|---|---|---|
| Full Fine-tuning | 62.24 | 87.18 | 91.30 | 92.33 | 92.28 | 79.42 | 94.57 | 90.92 | 86.28 |
| AccLoRT | **62.08** | 86.02 | **93.36** | 91.82 | 90.97 | **80.14** | 94.15 | 90.78 | **86.17** |
| GaLore (rank 8) | 60.60 | **87.17** | 92.01 | 92.2 | 91.11 | 79.78 | **94.38** | **90.82** | 85.94 |
| LoRA (rank 8) | 61.83 | 86.94 | 91.90 | **92.25** | **91.22** | 79.06 | 93.46 | 90.80 | 85.93 |

### 5.6 SCALING TO 7B LLaMA MODEL

To further demonstrate the memory efficiency of AccLoRT, in this part, we scale the pretraining task on a 7B LLaMA model. We conduct the experiments on 8 NVIDIA A100 40GB GPUs. We train the model for 10k steps with a batch size of 512 and set rank=512. The results are shown in Figure 8. We can see that AccLoRT achieves very stable training and evaluation performance under different accumulation frequencies, indicating its effectiveness and efficiency.

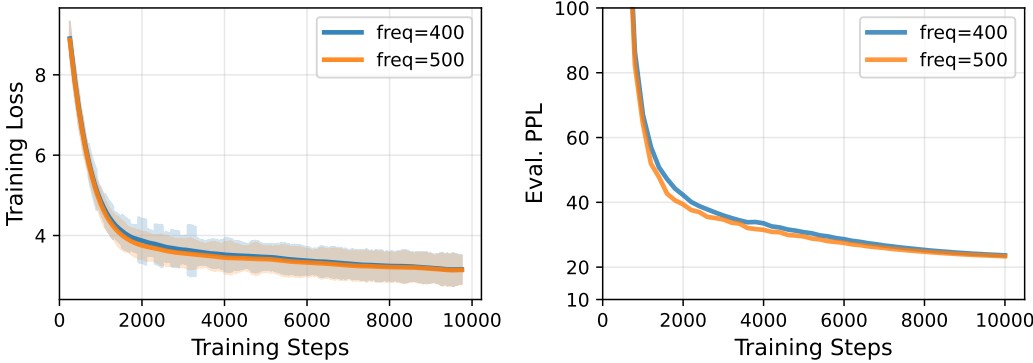

Figure 8: Comparison of training loss curves and evaluation PPL curves between different accumulation frequency for training a 7B model.

## 6 CONCLUSION

In this work, we present AccLoRT, a novel approach that enables memory-efficient pre-training of LLMs through progressive accumulation of low-rank adaptations. Our theoretical analysis establishes the rank evolution dynamics of the method and proves its equivalence to full-rank training after finite accumulation steps. Through extensive experiments on Llama models ranging from 60M to 1B parameters, we demonstrate that AccLoRT achieves competitive or superior performance compared to existing methods while maintaining consistent memory efficiency throughout the entire training process. The success of our low-rank training paradigm suggests a promising direction for making LLM development more accessible and efficient. Building on AccLoRT's memory-efficient training pipeline, more future work, including extending to the other transformer-based backbones such as ViT, can be further investigated.

## ETHICS STATEMENT

This work uses only computational methods and publicly available datasets, with no human subjects or private data. It follows the ICLR Code of Ethics, with no conflicts of interest. While acknowledging potential dual-use concerns, we stress responsible deployment and adhere to research integrity. All methods and results are reported transparently to support reproducibility.

## REPRODUCIBILITY STATEMENT

We provide code in the link and illustrative implementation details in the appendix to support reproduction of the main results.

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

## A  STATEMENT OF THE USE OF LARGE LANGUAGE MODELS (LLMS)

In this paper, we just used the LLM, ChatGPT, to polish the language of the paper. We did not use LLMs to generate any content or ideas in this work. We have verified the accuracy of all content and ideas in the paper.

## B  ADDITIONAL EXPERIMENTAL RESULTS

### B.1  TRAINING DYNAMICS COMPARISON

For a direct comparison of training dynamics, we benchmarked AccLoRT against GaLore on the LLaMA 60M model in Figure 9. As shown in the training Loss plot (left), the convergence curves for AccLoRT and GaLore are nearly identical. For evaluation perplexity (right), both methods perform similarly in the initial phase. However, after 4000 steps, AccLoRT consistently achieves a lower perplexity, leading to superior final performance. This experiment demonstrates that AccLoRT achieves excellent generalization while maintaining a competitive memory efficiency.

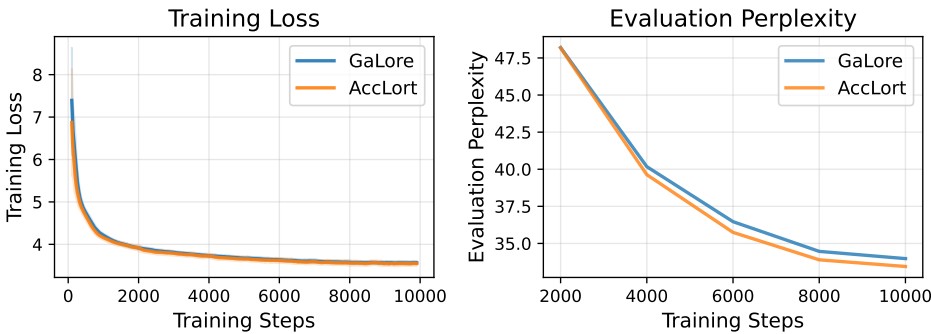

Figure 9: Training dynamics comparison between AccLoRT and GaLore on the LLaMA 60M model. The left plot (Training Loss) shows that both methods have nearly identical convergence curves. The right plot (Evaluation Perplexity) shows that while the methods perform similarly in the initial phase, AccLoRT consistently achieves a lower perplexity than GaLore after approximately 4000 training steps.

### B.2  FREQUENCY SELECTION AND FURTHER ABLATION

Regarding rank selection, a practical heuristic is to adapt the rank to the available memory constraints, which in turn determines the accumulation frequency. In our pretraining experiments, we typically set the rank to approximately 10% of the average matrix dimension. Consequently, the accumulation frequency is calibrated based on the proportion of full-parameter training steps required to achieve full-rank coverage. Formally, we define this relationship as:

$$T = \frac{t_{\text{total\_iter}}}{n/r} \times C, \tag{1}$$

where $t_{\text{total\_iter}}$ is the number of full-parameter training steps, $n$ is the average matrix size, $r$ is the rank, and $C \in [0.6, 1]$ is a scaling constant depending on the target convergence behavior.

We conducted an ablation study to analyze the AccLoRT's sensitivity to the frequency parameter $T$ on the LLaMA 60M model. According to (1), for 60M model, when giving $r = 100$, the frequency $T \in [1000, 2000]$ and giving $r = 128$, the frequency $T \in [1500, 2500]$. We tested five different frequencies, ranging in the interval of $[1000, 3000]$, using two different model ranks.

As shown in these Figures 10 and 11, the frequency parameter has a negligible impact on the model's performance. For both ranks, the training loss curves (left plots) are almost indistinguishable. The evaluation PPL curves (right plots) show minor variations during the early-to-mid training phase, but all lines ultimately converge to the same perplexity value by 10000 steps. This suggests that the model is highly robust to the frequency settings.

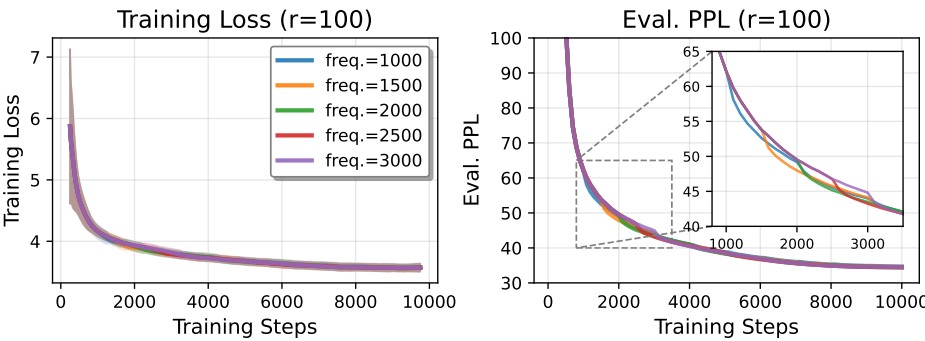

Figure 10: Impact of the update frequency (freq.) on training loss and evaluation PPL for rank $r = 100$. The plots compare different frequencies from 1000 to 3000. Both training loss and evaluation PPL curves show nearly identical convergence behavior, indicating performance is robust to the freq. parameter at this rank.

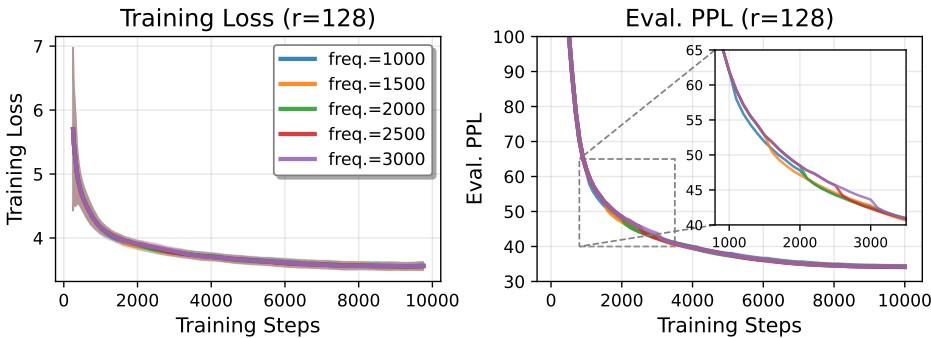

Figure 11: Impact of the update frequency (freq.) on training loss and evaluation PPL for rank $r = 128$. Similar to the results for $r = 100$, all tested freq. values yield nearly overlapping training loss curves. The evaluation PPL curves also converge to the same final value, confirming that the model's performance is not sensitive to the choice of freq. in the 1000-3000 range.

## B.3 INVESTIGATION ON EFFECTIVENESS OF RANK IN VARYING TRAINING STAGE

This ablation study highlights the impact of rank on model efficiency and representation capacity. As shown in Figure 12, the curves consistently shift downward with increasing training steps, indicating overall improvements in perplexity as the model trains longer. In addition, the relative advantage of higher ranks is maintained across all stages, suggesting that additional rank capacity is effectively utilized throughout training. Notably, the gap between ranks 25 and 100 remains substantial even at later steps, while the incremental benefits from increasing rank beyond 100 gradually taper off. This pattern suggests that moderate rank settings (e.g., 100) strike a favorable balance between parameter and memory efficiency and performance gains, whereas higher ranks (e.g., 200) provide diminishing returns.

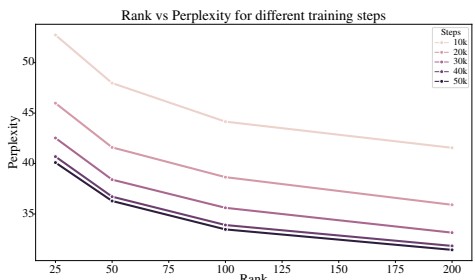

Figure 12: Performance of a Llama 60m in function of the rank at different training stages.

B.4  INVESTIGATION ON DIFFERENT FREQUENCIES AND RANKS

We further illustrate the experimental results for Section 5.1. Figure 13 provides a deeper look into how accumulation frequency and rank affect the reconstruction of singular values during training. On the left, with rank 8 and an accumulation frequency of 500, the reconstructed spectrum exhibits abrupt truncations after every multiple of rank, leaving many singular directions unrecovered. This reflects the inherent limitation of the low-rank approximation. As the training steps increase, the reconstructed spectrum gradually approaches the standard training. In the right panel, we show the evolution of singular values under standard linear regression with SGD optimization, where the spectrum is progressively filled in a smooth and continuous manner, eventually approximating the full distribution without sharp collapses.

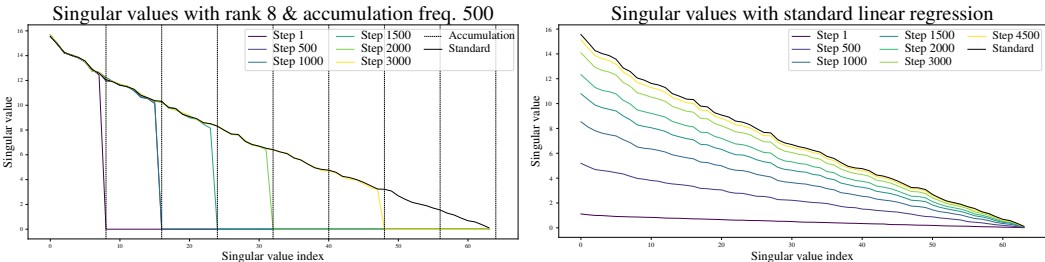

Figure 13: Reconstruction of singular values throughout the training of AccLoRT linear regression for different accumulation frequencies and ranks vs standard linear regression (SGD optimization)

B.5  ADAPTIVE RANK IN EARLY STAGES

In the early training stage, AccLoRT requires only $\mathcal{O}(dr(m+n))$ memory for the low-rank matrices $A_t$ and $B_t$, while GaLoRe needs $\mathcal{O}(mn)$ memory for the full weight matrix. This significant memory saving allows for potential performance improvements if we adaptively adjust the rank of AccLoRT during training. In detail, according to Table 1, the rank for AccLoRT in different accumulation stages $d$ can be given as

$$r_d = \max\left(r + \frac{mn}{3(m+n)} - \frac{\sum_{i=0}^{d-1} r_i}{3}, r\right), \tag{2}$$

where $r$ denotes the pregiven rank. Therefore, to maintain consistent memory usage, we can let the rank of AccLoRT to be relatively large in the early stage, which can further improve the performance of AccLoRT. In the following, we present the preliminary performance of AccLoRT with the above adaptively rank strategy for pretraining LLaMA 60M model. We can observe that with adaptive rank in the early stage, the AccLoRT can be further improved. From Table 5, we can see that AccLoRT with adaptive rank achieves a perplexity of 33.34, outperforming the standard AccLoRT's perplexity of 33.77. This result indicates that dynamically adjusting the rank during training can lead to better model performance while still benefiting from the memory efficiency of the AccLoRT framework.

Table 5: Perplexity comparison of AccLoRT with and without adaptive rank strategy on LLaMA 60M model.

| Method | 60M LLaMA |
|---|---|
| AccLoRT | 33.77 |
| AccLoRT (w. adaptive) | **33.34** |

B.6  ADDITIONAL FINE-TUNING EVALUATION ON GSM8K

To further evaluate AccLoRT in the fine-tuning task, we conducted additional fine-tuning experiments on the GSM8K dataset, a challenging benchmark for mathematical reasoning. We utilized the Mistral-7B model with a fixed rank of $r = 8$ and a batch size of 4.

**Superior Performance on Reasoning Tasks.**    We compare the performance of AccLoRT against standard baselines, including LoRA with the same rank. As shown in Table 6, AccLoRT significantly outperforms LoRA. This result validates our hypothesis that AccLoRT's ability to access a higher effective rank through subspace accumulation is particularly advantageous for complex mathematical reasoning tasks.

**Robustness to Frequency Selection.**    A key concern in AccLoRT is the sensitivity to frequency selection, particularly the accumulation frequency $T$. To investigate this, we performed a sweep of the update frequency across a wide range ($T \in [100, 800]$) while keeping the rank fixed at $r = 8$. The results, presented in Table 7, reveal a remarkably flat performance trajectory. The accuracy remains stable regardless of the specific frequency chosen. This indicates that once a reasonable rank is selected, the accumulation frequency acts as a second-order hyperparameter, allowing users to select mid-range values (e.g., $T \in [400, 700]$) without the need for extensive per-task tuning.

Table 6: Performance comparison on GSM8K (Mistral-7B). We compare AccLoRT with LoRA under the same trainable parameter budget ($r = 8$, batch size=4).

| Method | Accuracy (%) |
|---|---|
| LoRA | 47.84 |
| AccLoRT ($T = 600$) | **54.44** |

Table 7: Sensitivity analysis of accumulation frequency on GSM8K. Using Mistral-7B with a fixed rank of $r = 8$, we sweep the accumulation frequency $T$ from 100 to 800.

| Frequency ($T$) | Exact Match Acc. (%) |
|---|---|
| 100 | 53.22 |
| 200 | 53.68 |
| 250 | 52.99 |
| 300 | 51.48 |
| 400 | 52.08 |
| 500 | 53.68 |
| 600 | **54.44** |
| 700 | 54.28 |
| 800 | 53.83 |

### B.7    COMPUTATIONAL OVERHEAD ON ACCUMULATION

In this section, we evaluate the practical efficiency of AccLoRT by measuring the total wall-clock training time. We conducted experiments using the LLaMA 60M model on a single NVIDIA A100 (40GB) GPU. We note that direct comparisons with full-parameter training regarding wall-clock time and power consumption were not feasible on this specific low-resource setup, as full-parameter training triggers OOM errors. To quantify the computational overhead introduced by the accumulation steps in AccLoRT, we measured the training time across varying settings of rank ($r$) and accumulation frequency ($T$). The results are presented in Table 8.

We can observe that the overhead introduced by the method is minimal. Even when using a high accumulation frequency (e.g., $T = 1000$), the total training time increases only marginally compared to infrequent accumulation settings (e.g., $T = 5000$). This demonstrates that AccLoRT does not constitute a significant computational bottleneck, making it efficient for practical pre-training scenarios.

## C    EXPERIMENTAL SETTINGS IN PRE-TRAINING

**Architecture parameters**    In all of our experiments, we used Llama model architecture (Touvron et al., 2023a;b) from the HuggingFace library (Wolf et al., 2020), and adopt the same experimental

Table 8: Total wall-clock training time of AccLoRT on LLaMA 60M under different ranks $r$ and accumulation frequencies $T$. Experiments were conducted on a single NVIDIA A100 (40GB) GPU.

| Frequency ($T$) | Rank ($r$) | | | |
|---|---|---|---|---|
| | 50 | 100 | 150 | 200 |
| 1000 | 2h 13m | 2h 17m | 2h 21m | 2h 17m |
| 2000 | 2h 6m | 2h 10m | 2h 14m | 2h 9m |
| 3000 | 2h 2m | 2h 6m | 2h 10m | 2h 6m |
| 4000 | 2h 2m | 2h 6m | 2h 10m | 2h 6m |
| 5000 | 2h 2m | 2h 6m | 2h 10m | 2h 5m |

setting as in Zhao et al. (2024). The main parameters such as the hidden size, the number of parameters and number of heads of each architecture are listed in Table 9. The data amount represents the theoretical number of tokens that needs to be fed to the transformer to achieve reasonable convergence. This results is known as the Chinchilla scaling law of large language model (Hoffmann et al., 2022) and states that one needs about 17 times the number of parameters in terms of tokens.

Table 9: Architecture of the Llama models that were used for the experiments.

| Params | Hidden | Intermediate | Heads | Layers | Seq. len. | Data amount |
|---|---|---|---|---|---|---|
| 60M | 512 | 1376 | 8 | 8 | 256 | 1.3B |
| 130M | 768 | 2048 | 12 | 12 | 256 | 2.6B |
| 350M | 1024 | 2736 | 16 | 24 | 256 | 7.8B |
| 1.3B | 2048 | 5461 | 32 | 24 | 256 | 13.1B |

**Training hyperparameters** For each Llama model using the AccLoRT method presented in Table 3, their hyperparameters can be referred to in Table 10. The memory usage for all baselines is given in Table 11.

Table 10: Training hyperparameters of the Llama models that were used for the experiments (the batchsize is per GPU).

| Params | Batchsize | Training steps | Warmup steps | Optimizer | Learning rate | rank | AccLoRT frequency |
|---|---|---|---|---|---|---|---|
| 60M | 512 | 10 000 | 5% | AdamW | 2.5e-3 | 100 | 2000 |
| 130M | 512 | 20 000 | 5% | AdamW | 2e-3 | 200 | 5000 |
| 350M | 512 | 60 000 | 5% | AdamW | 5e-4 | 200 | 5000 |
| 1.3B | 512 | 100 000 | 5% | AdamW | 5e-4 | 300 | 5000 |

# D EXPERIMENTAL SETTINGS IN FINE-TUNING

**GLUE tasks** To assess the quality of AccLoRT for finetuning on top of pretraining, we used two widely used datasets: the GLUE dataset Wang et al. (2019) comprising multiple tasks on natural language understanding (NLI) such as CoLA, MNLI, MRPC, QNLI, QQP, RTE, SST2 and STS-B on which we tuned a RoBERTa-base model. Table 12 lists the hyperparameters used for the tuning of a RoBERTa-base model on the different tasks of GLUE.

**Smoothing the parameter surfaces** Altough it is known that the finetuning of pretrained model only needs to occur in a small subspace of the model parameter space (see Hu et al. (2021); Gur-Ari et al. (2018); Larsen et al. (2022); Aghajanyan et al. (2020) for thoughts on low rank in deep learning finetuning and pretraining), it is not clearly obvious which subspace a low-rank adapter would optimize. Experimentally, it is to be observed that there is no pattern and that each adapter will chose a subspace corresponding to its rank and update on this one depending on the first few training steps when the rank of the zero initialized matrix becomes full. While the accumulation process of

Table 11: Model size for weights+gradients and optimizer states in GB. The memory usage of AccLoRT increases with accumulation. We report both the lower and upper bounds.

| Llama | Modules | 60M | 130M | 350M | 1B |
|---|---|---|---|---|---|
| Full-rank | weights + gradients | 0.23G | 0.54G | 1.47G | 5.36G |
| | optimizer | 0.23G | 0.54G | 1.47G | 5.36G |
| LoRA | weights + gradients | 0.16G | 0.38G | 0.74G | 2.44G |
| | optimizer | 0.16G | 0.38G | 0.74G | 2.44G |
| | $r/d$ | 128 / 512 | 256 / 768 | 256 / 1024 | 512 / 2048 |
| ReLoRA | weights + gradients | 0.20G | 0.46G | 1.11G | 3.90G |
| | optimizer | 0.16G | 0.38G | 0.74G | 2.44G |
| | $r/d$ | 128 / 512 | 256 / 768 | 256 / 1024 | 512 / 2048 |
| GaLore | weights + gradients | 0.23G | 0.54G | 1.47G | 5.36G |
| | optimizer | 0.16G | 0.38G | 0.74G | 2.44G |
| | $r/d$ | 128 / 512 | 256 / 768 | 256 / 1024 | 512 / 2048 |
| SLTrain | weights + gradients | 0.17G | 0.40G | 0.85G | 2.87G |
| | optimizer | 0.17G | 0.39G | 0.78G | 2.08G |
| | $r/d$ | 128 / 512 | 256 / 768 | 256 / 1024 | 512 / 2048 |
| AccLoRT | weights + gradients | $0.13 \sim 0.18$G | $0.30 \sim 0.42$G | $0.60 \sim 1.02$G | $1.43 \sim 3.40$G |
| | optimizer | 0.13G | 0.30G | 0.60G | 1.43G |
| | $r/d$/frequency | 100 / 512 / 2000 | 200 / 768 / 5000 | 200 / 1024 / 5000 | 300 / 2048 / 5000 |

Table 12: Hyperparameters used for the fine-tuning of RoBERTa-base models using SoW on the GLUE dataset.

| | CoLA | MNLI | MRPC | QNLI | QQP | RTE | SST2 | STS-B |
|---|---|---|---|---|---|---|---|---|
| EPOCHS | 30 | 30 | 30 | 30 | 30 | 30 | 30 | 30 |
| BATCH SIZE | 16 | 16 | 16 | 16 | 16 | 16 | 16 | 16 |
| LEARNING RATE | 1.5E-6 | 1E-6 | 1E-5 | 1.5E-6 | 2E-6 | 1E-6 | 1.5E-6 | 5E-6 |
| SoW LEARNING RATE | 7.5E-6 | 7.5E-5 | 5E-4 | 7.5E-5 | 1.2E-4 | 1.2E-4 | 7.5E-5 | 1E-5 |
| RANK | 10 | 20 | 5 | 10 | 20 | 20 | 10 | 20 |
| ACCUMULATION FREQ. | 7 500 | 10 000 | 1 500 | 7 500 | 100 000 | 2 500 | 7 500 | 2 500 |
| TRAINING SAMPLES | 8.5K | 393K | 3.7K | 105K | 364K | 2.5K | 67K | 5.7K |

AccLoRT enables to redraw the subspace onto which one adapter would learn, for finetuning one dataset may need to keep the most global structure of the weights and only focus on the dimension of smallest singular values. The easiest way to guide the adapters to do so is to remove those small singular valued subspaces from the weight. This can be done through a truncated SVD (or QR decomposition) on the weight parameters. Algorithm 2 illustrates this point where we perform a truncated QR and set the removed rows and columns from Q and R to be the A and B matrices of the adapter.

---

**Algorithm 2** AccLoRT algorithm by using QR decomposition

---

**Input:** $r$ the rank, $m$ additional space discarded $W \in \mathbb{R}^{n \times m}$ a weight matrix.
$Q, R \leftarrow QR(W)$          ▷ QR decomposition of $W$
$Q_{\text{major}} \leftarrow Q[:, : n - r - m]$          ▷ Keep the first $n - r$ columns of $Q$
$R_{\text{major}} \leftarrow R[: n - r - m, :]$          ▷ Keep the first $n - r$ columns of $R$
$Q_{\text{minor}} \leftarrow Q[:, -r :]$          ▷ Keep the last $r$ columns of $Q$
$R_{\text{minor}} \leftarrow R[-r :, :]$          ▷ Keep the last $r$ rows of $R$

$W_{\text{acc}} \leftarrow Q_{\text{major}} R_{\text{major}}$          ▷ Accumulate the major part
$A \leftarrow Q_{\text{minor}}$
$B \leftarrow R_{\text{minor}}$
**Output:** $W_{\text{acc}}, A, B$

---

While we do not provide experiments on this idea, we give some intuitions on it. First is that this method generally does not provide any improvement compared to traditionnal finetuning or standard AccLoRT finetuning but highly depends on the dataset. For some datasets, where we make the assumption that they may already perform well on zero-shot, this method gives better performances. Additionnaly, it could be interesting to perform a kind of subspace selection where we would run, before the training process, evaluations with several subspaces removed at random or with some programmed heuristics and take the best one as the decomposed model to further improve the learning capability of the finetuning process. Lastly, one could make a mix between setting both matrices A and B to be equal to the left and right singular vectors corresponding to the selected subspace where we want to learn and random initialisation of some rows/columns to select one subspace but let backpropagation select other subspaces to optimize.

## E    RANK OF SUM OF MATRICES

This annex section will present a few results on the rank of the sum of matrices. Expressely, we will remind the work of Marsaglla (1964) on the bounds of the rank of the sum of two matrices. We provide a generalization of these inequalities for a finite sum of $n$ matrices.

These bounds are important in that they allow to control the rank of the sum under several conditions that will be developed for random matrices in the next section.

**Lemma E.1.** *Let $A, B \in \mathbb{R}^{n \times m}$ 2 matrices of the same size.*
*Let $R_A = \text{span}\,(\text{row}\,(A))$, $R_B = \text{span}\,(\text{row}\,(B))$ be the spaces spanned by the rows of A and B. Let $C_A = \text{span}\,(\text{col}\,(A))$, $C_B = \text{span}\,(\text{col}\,(B))$ be the spaces spanned by the columns of A and B.*
*We have the two following equalities:*

*i.* $\dim\,(C_A + C_B) = \text{rank}\,([A \quad B])$

*ii.* $\dim\,(R_A + R_B) = \text{rank}\left(\begin{bmatrix} A \\ B \end{bmatrix}\right)$

*Proof.* We will show a more general result; that is $C_A + C_B = \text{Im}\,([A \quad B])$. Let $a_i, b_i$ for $1 \le i \le m$ be the columns of $A$ and $B$. Then, for any $v \in \text{Im}\,([A \quad B])$, there exists $v_i^A, v_i^B$ such that $v = \sum_{i=1}^m v_i^A a_i + v_i^B b_i$ and thus $v \in C_A + C_B$. Conversely, let $v \in C_A + C_B = \{xA + yB, x, y \in \mathbb{R}^n\}$. Then, there exists $x, y$ such that $v = xA + yB = [A \quad B] \begin{bmatrix} x \\ y \end{bmatrix}$ and thus $v \in \text{Im}\,([A \quad B])$. The proof for the rows is similar. $\qquad\square$

The notion of row (resp. column) dependency will be of great interest to characterize how the rowspace (resp. colspace) of two matrices interact. Indeed, in the following propositions, the bounds on the rank of the sum of multiple matrices will be greatly depending on the dimension of the intersection of their rowspace (resp. colspace). The row and column independency can be treated as whether the rowspaces (resp. colspaces) of two matrices are in direct sum but also how the rank of the block matrix made by the concatenation along the correct axis equalizes the sum of the ranks of both matrices.

**Definition E.2.** Let $n, m$ be intergers. We say that two matrices $A, B \in \mathbb{R}^{n \times m}$ are **row independent** noted $A \perp_R B$ (resp. **column independent** noted $A \perp_C B$) if $R_A \cap R_B = \{0\}$ (resp. $C_A \cap C_B = \{0\}$). They are said to be **independent** noted $A \perp B$ if they are both row and column independent.

**Proposition E.3.** *Let $A, B \in \mathbb{R}^{n \times m}$ 2 matrices of the same size. We have the following equivalences for the rows of A and B:*

*i. $A, B$ are row independent*

*ii. $R_A \cap R_B = \{0\}$*

*iii. $R_A + R_B = R_A \oplus R_B$*

*iv. $\dim\,(R_A + R_B) = \text{rank}\left(\begin{bmatrix} A \\ B \end{bmatrix}\right) = \text{rank}(A) + \text{rank}(B)$*

*and for the columns of A and B:*

*i. $A, B$ are column independent*

*ii. $C_A \cap C_B = \{0\}$*

*iii.* $C_A + C_B = C_A \oplus C_B$

*iv.* $\dim(C_A + C_B) = \mathrm{rank}([A \quad B]) = \mathrm{rank}(A) + \mathrm{rank}(B)$

*Proof.* We have for both the rows and the columns:

i. $\iff$ ii. by definition.

ii. $\iff$ iii. by definition of the direct sum.

ii $\iff$ iv. by the Lemma E.1 and $\dim(E + F) = \dim(E) + \dim(F) \iff E \oplus F$. $\qquad\square$

The next proposition expresses the upper and lower bound of the rank of $A + B$ for two matrices as given by George Marsaglla and rewrites it in terms of block matrices which will be of great use for our work on the generalization to any finite sum.

**Proposition E.4** (Inequalities for $\mathrm{rank}(A + B)$, Marsaglla (1964))**.** *Let $A, B$ be two matrices of the same size. Then,*

*i.* $\mathrm{rank}(A) + \mathrm{rank}(B) - \dim(R_A \cap R_B) - \dim(C_A \cap C_B) \leq \mathrm{rank}(A + B)$

*ii.* $\mathrm{rank}(A + B) \leq \mathrm{rank}(A) + \mathrm{rank}(B) - \max(\dim(R_A \cap R_B), \dim(C_A \cap C_B))$

*These inequalities can also be rewritten:*

*i.* $\mathrm{rank}([A \quad B]) + \mathrm{rank}\left(\begin{bmatrix} A \\ B \end{bmatrix}\right) - \mathrm{rank}(A) - \mathrm{rank}(B) \leq \mathrm{rank}(A + B)$

*ii.* $\mathrm{rank}(A + B) \leq \min\left(\mathrm{rank}([A \quad B]), \mathrm{rank}\left(\begin{bmatrix} A \\ B \end{bmatrix}\right)\right)$

*which can be derived using $\dim(E + F) = \dim(E) + \dim(F) - \dim(E \cap F)$ and the Lemma E.1.*

*Proof.* See Marsaglla (1964) for the first result. The rewritten inequalities follow from Lemma E.1 and the fact that $\mathrm{rank}(A) = \dim(R_A) = \dim(C_A)$, as

$$\dim(R_A \cap R_B) = \dim(R_A) + \dim(R_B) - \dim(R_A + R_B)$$
$$= \mathrm{rank}(A) + \mathrm{rank}(B) - \mathrm{rank}([A \quad B])$$

$$\dim(C_A \cap C_B) = \dim(C_A) + \dim(C_B) - \dim(C_A + C_B)$$
$$= \mathrm{rank}(A) + \mathrm{rank}(B) - \mathrm{rank}\left(\begin{bmatrix} A \\ B \end{bmatrix}\right)$$

$\qquad\square$

This double inequality provides a powerful way to bound the rank of the sum of two matrices. In our work, we are interested in controling the bounds of the rank of a finite number of matrices. The two following propositions give a generalization of the previous bounds for $d$ matrices.

**Lemma E.5.** *Let $A$ be a $n \times m$ matrix. Then the rank of $A$ is the rank of the projectors on the columns $P_{col} = A(A^\top A)^{-1} A^\top$ and on the rows $P_{row} = A^\top (AA^\top)^{-1} A$.*

$$\mathrm{rank}(A) = \mathrm{rank}(P_{row}) = \mathrm{rank}(P_{col})$$

*Proof.* By construction, $P_{\mathrm{row}}$ projects any vector into the row-space of $A$ that is $\mathrm{row}(A)$. Hence, $\mathrm{rank}(P) = \dim(\mathrm{Im}(P)) = \dim(\mathrm{row}(A)) = \mathrm{rank}(A)$. Similarly, we have $P_{\mathrm{col}} = \dim(\mathrm{Im}(P_{\mathrm{col}})) = \dim(\mathrm{col}(A)) = \mathrm{rank}(A)$. Where the last equality for both expressions comes from the fundamental theorem of linear algebra. $\qquad\square$

**Proposition E.6** (Upper bound)**.** *Let $(A_i)_{1 \leq i \leq d}$ be a finite family of matrices in $\mathbb{R}^{m \times n}$. Then, we have the upper bound for the rank of their sum:*

$$\mathrm{rank}\left(\sum_{i=1}^{d} A_i\right) \leq \min\left(\mathrm{rank}([A_1 \quad \cdots \quad A_d]), \mathrm{rank}\left(\begin{bmatrix} A_1 \\ \vdots \\ A_d \end{bmatrix}\right)\right)$$

*Proof.* Let $X = [A_1 \quad \cdots \quad A_d]$ and $Y = [P_1 \quad \cdots \quad P_d]^\top$ where, for all $1 \leq i \leq d$, $P_i$ is the orthogonal projector on the row spaces of the $A_i$. Then, we can bound the rank of the product of $XY$ as $\mathrm{rank}(XY) \leq \min(\mathrm{rank}(X), \mathrm{rank}(Y))$.

By construction, $\mathrm{rank}(X) = \mathrm{rank}\left([A_1 \quad \cdots \quad A_d]\right)$ and $\mathrm{rank}(Y) = \mathrm{rank}\left(\begin{bmatrix} A_1 \\ \vdots \\ A_d \end{bmatrix}\right)$ so that we

have the desired result recalling that $\sum_{i=1}^n A_i = XY$. $\qquad\qquad\square$

**Proposition E.7** (Lower bound). *Let $(A_i)_{1 \leq i \leq d}$ be a finite family of matrices in $\mathbb{R}^{m \times n}$. Then, we have the lower bound for the rank of their sum:*

$$\mathrm{rank}\left([A_1 \quad \cdots \quad A_d]\right) + \mathrm{rank}\left(\begin{bmatrix} A_1 \\ \vdots \\ A_d \end{bmatrix}\right) - \sum_{i=1}^d \mathrm{rank}\left(A_i\right) \leq \mathrm{rank}\left(\sum_{i=1}^d A_i\right)$$

*Proof.* The following follows and details the sketch of proof given by Ozawa (2024).

Let $X = [A_1 \quad \cdots \quad A_d]$ and $Y = [P_1 \quad \cdots \quad P_d]^\top$ where, for all $1 \leq i \leq d$, $P_i$ is the orthogonal projector on the row spaces of the $A_i$.
From the rank-nullity theorem, we have $\mathrm{rank}(Y) + \dim(\mathrm{Ker}(Y)) = n$ and $\mathrm{rank}(XY) + \dim(\mathrm{Ker}(XY)) = n$.
Observe that any vector in $\mathrm{Ker}(XY)$ comes either from $\mathrm{Ker}(Y)$ (as $\mathrm{Ker}(Y) \subset \mathrm{Ker}(XY)$) or from elements of $\mathrm{Im}(Y)$ mapped to 0 by $X$ i.e elements of $\mathrm{Im}(Y) \cap \mathrm{Ker}(X)$. Hence, $\mathrm{Ker}(XY) = \mathrm{Ker}(Y) \oplus (\mathrm{Im}(Y) \cap \mathrm{Ker}(X))$ and $\mathrm{rank}(XY) = n - \dim(\mathrm{Ker}(XY)) = n - \dim(\mathrm{Ker}(Y)) - \dim(\mathrm{Im}(Y) \cap \mathrm{Ker}(X)) = \mathrm{rank}(Y) - \dim(\mathrm{Im}(Y) \cap \mathrm{Ker}(X))$. Now, as $\mathrm{Ker}(X)$ is in direct sum with $(\mathrm{Ker}(X))^\perp$ by definition and $\mathrm{Ker}(X) \cap \mathrm{Im}(Y) \subset \mathrm{Ker}(X)$, we have that $(\mathrm{Ker}(X) \cap \mathrm{Im}(Y)) \oplus (\mathrm{Ker}(X))^\perp$. Hence,

$$\dim(\mathrm{Im}(Y) \cap \mathrm{Ker}(X)) + \dim((\mathrm{Ker}(X))^\perp) = \dim(\mathrm{Im}(Y) \cap \mathrm{Ker}(X) + (\mathrm{Ker}(X))^\perp)$$

and as $(\mathrm{Ker}(X)^\perp)$ is isomorphic to $\mathrm{Im}(X)$ by the first isomorphism theorem, $\dim((\mathrm{Ker}(X))^\perp) = \mathrm{rank}(X)$.
Finally, we can deduct that,

$$\mathrm{rank}(XY) = \mathrm{rank}(Y) - \dim(\mathrm{Im}(Y) \cap \mathrm{Ker}(X))$$
$$= \mathrm{rank}(Y) + \mathrm{rank}(X) - \dim((\mathrm{Im}(Y) \cap \mathrm{Ker}(X)) \oplus (\mathrm{Ker}(X))^\perp)$$

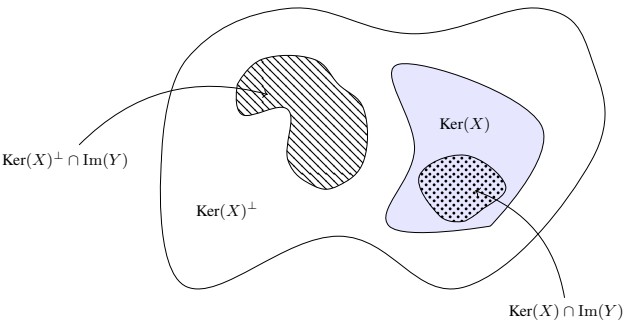

Figure 14: Decomposition of the space for $\mathrm{Ker}(X)$ and its complementary along with the image of $Y$.

Let us work with the space $\mathrm{Im}(Y) + (\mathrm{Ker}(X))^\perp \supset (\mathrm{Im}(Y) \cap \mathrm{Ker}(X)) \oplus (\mathrm{Ker}(X))^\perp$. For any $i \in \{1, \ldots, d\}$, $\mathrm{Im}(P_i) = \mathrm{Im}(A_i^\top) = \mathrm{Ker}(A_i)^\perp$. The last expression is true as for any $v \in \mathrm{Im}(A_i^\top)$, there exists a vector $w$ such that $v = A_i^\top w$ and for any $u \in \mathrm{Ker}(A_i)$, $A_i^\top w \cdot u = w^\top \cdot A_i u = 0$ meaning that $v \in (\mathrm{Ker}(A_i))^\perp$ (the other inclusion is proved similarly). We can

conclude that $\mathrm{Im}(Y) \subset (\mathrm{Ker}(A_1))^\perp + \cdots + (\mathrm{Ker}(A_d))^\perp$ as any $y \in \mathrm{Im}(Y)$ can be written from $x = \begin{bmatrix} x_1^\top & \cdots & x_d^\top \end{bmatrix}^\top$ as $y = \sum_{i=1}^d P_i x_i = \sum_{i=1}^d y_i$ with $y_i = P_i x_i \in \mathrm{Im}(P_i)$ for all $1 \le i \le d$ by construction of $Y$.

This results in $\mathrm{Im}(Y) + (\mathrm{Ker}(X))^\perp \subset (\mathrm{Ker}(A_1))^\perp + \cdots + (\mathrm{Ker}(A_d))^\perp$ and as $\dim\left((\mathrm{Ker}(A_1))^\perp + \cdots + (\mathrm{Ker}(A_d))^\perp\right) \le \sum_{i=1}^d \dim((\mathrm{Ker}(A_i))^\perp) = \sum_{i=1}^d \mathrm{rank}(A_i)$, we have that:

$$\mathrm{rank}\left(\sum_{i=1}^d A_i\right) = \mathrm{rank}(XY) \ge \mathrm{rank}(X) + \mathrm{rank}(Y) - \sum_{i=1}^d \mathrm{rank}(A_i)$$

By construction, $\mathrm{rank}(X) = \mathrm{rank}\left(\begin{bmatrix} A_1 & \cdots & A_d \end{bmatrix}\right)$ and $\mathrm{rank}(Y) = \mathrm{rank}\left(\begin{bmatrix} A_1 \\ \vdots \\ A_d \end{bmatrix}\right)$ so that we

have the desired inequality.

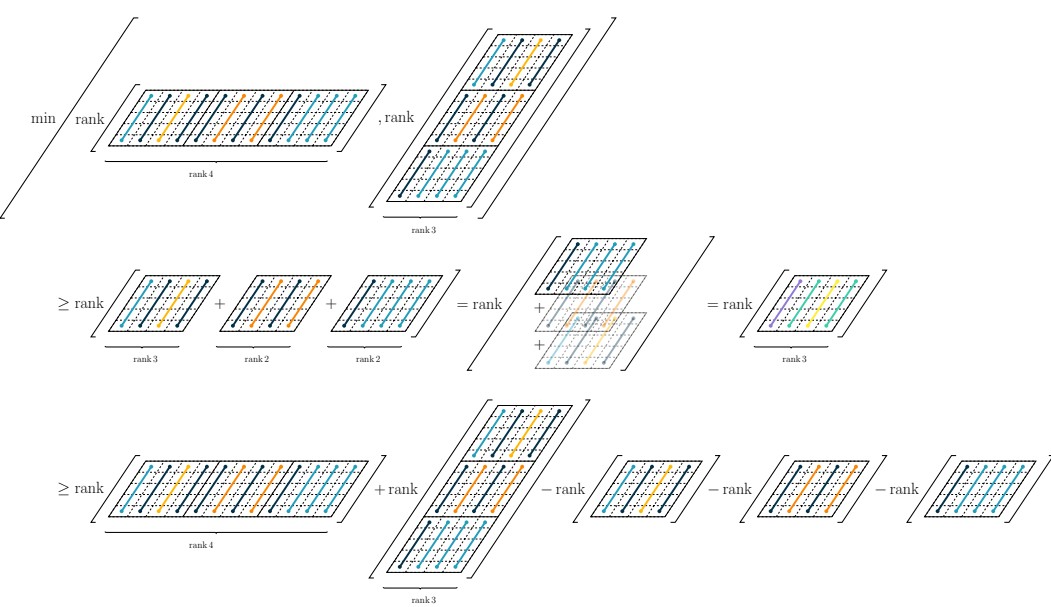

Figure 15: Illustration of the bound inequalities for the rank of three matrices ($3 \ge \mathrm{rank}(A + B + C) \ge 1$).

$\square$

## F  SOME RESULTS ON RANDOM VECTORS AND MATRICES

**Proposition F.1.** *The set $\Delta_k$ of all $k$-families of linearly dependent vectors of $\mathbb{R}^n$ has zero measure for all $0 \le k \le n$ for the Lebesgue measure or any absolutely continuous measure with respect to the Lebesgue measure.*

This proposition allows for the direct corollary that the set of singular matrices is of null measure under any absolutely continuous measure. This can be proven seeing that such set is $\{A \mid \det(A) = 0\}$ which is exactly $\Delta_n$ and that a determinant is zero if and only if the column of its matrix are not linearly independent. As such, any random matrix with i.i.d entries from an absolutely continuous probability (w.r.t the Lebesgue measure) is thus full-rank with probability one.

**Lemma F.2.** *Let $X$ be a set which has zero measure for the Lebesgue measure. Let $D$ be a discrete set of $\mathbb{R}$ with associated probability measure $\rho$ ($\rho(D) = 1$). Then, $X \times D$ has zero measure for the measure $\lambda \otimes \rho$.*

*Proof.* We can decompose the set as $X \times D = \left( \bigcup_{x \in D} (X \times \{x\}) \right)$. By definition of a null set, for $\varepsilon > 0$, there exists a countable sequence $(U_i)_i$ of open intervals for the underlying $\sigma$-algebra such that $X \subset \bigcup_i U_i$ and $\sum_i \text{length}(U_i) < \varepsilon$.

Thus, we have that $X \times \{x\} \subset (\bigcup_i U_i) \times \{x\}$ which has length $\leq \varepsilon$ as $\rho(\{x\}) \leq 1$. As the countable union of sets of measure zero is a set of measure zero, we can conclude that $X \times D$ has zero measure for the measure $\lambda \otimes \rho$. $\qquad\square$

**Proposition F.3.** *Let $D$ be a discrete set of vectors of $\mathbb{R}^n$ with an associated probability measure $\rho$ and $\Delta_k^* \subset (\mathbb{R}^n)^{k-1} \times D$ the set of $k$-families of linearly dependent vectors of $\mathbb{R}^n$ where the last one is drawn from $D$. The set $\Delta_k^*$ has null measure for all $0 \leq k \leq n$ for the measure $\lambda_{k-1} \times \rho$.*

*Proof.* For $k = 0$, any vector of $D$ is an independent family of 1 vector. For $k > 0$, the proof relies on the fact that

$$\Delta_k^* = (\Delta_k^* \cap (\Delta_{k-1} \times D)) \cup (\Delta_k^* \cap \overline{\Delta_{k-1} \times D})$$
$$= (\Delta_{k-1} \times D) \cup (\Delta_k^* \backslash (\Delta_{k-1} \times D))$$

as $\Delta_{k-1} \times D \subset \Delta_k^*$ (adding a vector to a dependent family still yields a dependent family). That is, this set is either formed from the already dependent $k-1$-families of $\mathbb{R}^n$ or the last element introduces dependency from the independent $k-1$-families of $\mathbb{R}^n$. This set is measurable by the union and product of measurable sets on $(\mathbb{R}^n \mathcal{B}(\mathbb{R}^n)$ and $(D, \mathcal{P}(D))$.

We have the equivalence,

$$(x_1, \ldots, x_{k-1}, x) \in \Delta_k^* \backslash (\Delta_{k-1} \times D) \iff (x_1, \ldots, x_{k-1}) \notin \Delta_{k-1} \text{ and } x \in \text{span}(\{x_1, \ldots, x_{k-1}\}))$$

Furthermore, from Lemma F.2 (the extension from $\mathbb{R}$ to $\mathbb{R}^n$ is straightforward), $(\lambda_{k-1} \otimes \rho)(\Delta_{k-1} \times D) = 0$ as $\lambda_{k-1}(\Delta_{k-1}) = 0$ from Proposition F.1.

Then, with Tonelli's theorem ($\lambda_{k-1}$ and $\rho$ are $\sigma$-finite measure and the integrand is positive),

$$(\lambda_{k-1} \otimes \rho)(\Delta_k^*) = (\lambda_{k-1} \otimes \rho)(\Delta_{k-1} \times D) + (\lambda_{k-1} \otimes \rho)(\Delta_k^* \backslash (\Delta_{k-1} \times D))$$
$$= (\lambda_{k-1} \otimes \rho)(\Delta_k^* \backslash (\Delta_{k-1} \times D))$$
$$= \int \mathbb{1}_{\text{span}(\{x_1, \ldots, x_{k-1}\})}(x) \mathbb{1}_{\overline{\Delta_{k-1}}}(x_1, \ldots, x_{k-1}) \, d\rho(x) \, d\lambda(x_1) \cdots d\lambda(x_{k-1})$$
$$= \int \left( \int \mathbb{1}_{\text{span}(\{x_1, \ldots, x_{k-1}\})}(x) d\rho(x) \right) \mathbb{1}_{\overline{\Delta_{k-1}}}(x_1, \ldots, x_{k-1}) \, d\lambda(x_1) \cdots d\lambda(x_{k-1})$$
$$= 0$$

where the last line comes from the fact that a polynomial from $\mathbb{R}^n$ to $\mathbb{R}$ is either zero or non-zero almost everywhere (see Caron (2005)) for the measure $\rho$ and $\text{span}(\{x_1, \ldots, x_{k-1}\})$ is a $k-1$-dimensional hyperplane $H$ such that there exists a non zero linear functional $\varphi$ where $H = \{x \in \mathbb{R}^n \mid \varphi(x)\}$ and $H$ passes through 0. However, the polynomial is a non zero polynomial by definition, hence, the measure of $H$ by $\rho$ is null. $\qquad\square$

**Lemma F.4.** *Let $n, m \geq 1$ and $r \leq n, m$. For two full-rank matrices $A \in \mathbb{R}^{n \times r}$ and $B \in \mathbb{R}^{r \times m}$, $\text{rank}(AB) = r$.*

*Proof.* From Sylvester's rank inequality, we have $\text{rank}(AB) \geq \text{rank}(A) + \text{rank}(B) - r = r$. But $\text{rank}(AB) \leq \min(\text{rank}(A), \text{rank}(B)) = r$. Thus, $\text{rank}(AB) = r$. $\qquad\square$

**Definition F.5.** Let $n, m, r$ be integers.
The set of all random matrices in $\mathbb{R}^{n \times m}$ with i.i.d entries under the probability distribution $P$ is denoted by $\mathbb{M}_{n,m}(P) = \{(a_{ij}) \in \mathbb{R}^{n \times m}, \ a_{ij} \sim P \ \text{i.i.d.}\}$.
The set of all random rank-$r$ matrices ($r \leq n, m$) in $\mathbb{R}^{n \times m}$ given by the product of two matrices in $\mathbb{R}^{n \times r}$ and $\mathbb{R}^{r \times m}$ with i.i.d entries under the probability distribution $P$ is denoted by

$$\mathbb{M}_{n,m}^r(P) = \{AB, \ (A, B) \in \mathbb{M}_{n,r}(P) \times \mathbb{M}_{r,m}(P)\}$$

Indeed, by Lemma F.4, the product of two random matrices has rank $\min(n, m, r)$ with probability 1.

In the following, we will note $\mathbb{M}_{n,m}^{r_i}$ the set $\mathbb{M}_{n,m}^{r_i}(P)$ with $P$ can be any absolutely continuous probability w.r.t the Lebesgue measure.

**Lemma F.6.** *Let $n, m, d$ be integers and $(r_i)_{1 \le i \le d}$ be a finite family of integers. For $1 \le i \le d$, let $A_i \in \mathbb{M}_{n,m}^{r_i}$. Then, with probability 1,*

$$\mathrm{rank}\,([A_1 \quad \cdots \quad A_d]) = \min\left(n, \sum_{i=1}^{d} r_i\right)$$

$$\mathrm{rank}\left(\left[A_1^\top \quad \cdots \quad A_d^\top\right]^\top\right) = \min\left(m, \sum_{i=1}^{d} r_i\right)$$

*Proof.* For each $A_i$ extract $r_i$ rows that form a linearly independent family. Let us note them $(a_1^i, \ldots, a_{r_i}^i)$. Thus, $\mathrm{rank}\,([A_1 \quad \cdots \quad A_d]) = \mathrm{rank}\left(\{a_1^1, \ldots, a_{r_1}^1, \ldots, a_1^d, \ldots, a_{r_d}^d\}\right)$ (where the right-hand side corresponds to the rank of the space spanned by these vectors). Suppose $\sum r_i \le n$, by Proposition F.1, this family is linearly independent with probability 1 as the distributions $\mathbb{M}_{n,m}^{r_i}$ are absolutely continuous and $\mathrm{rank}\left(\{a_1^1, \ldots, a_{r_1}^1, \ldots, a_1^d, \ldots, a_{r_d}^d\}\right) = \sum_{i=1}^{d} r_i$. If $\sum r_i > n$, we can remove enough vectors to fall back to the previous case. $\square$

**Theorem F.7.** *Let $n, m, d \ge 1$ be integers. For $i \in \{1, \ldots, d\}$, let $1 \le r_i \le \min(n, m)$ and $A_i \in \mathbb{M}_{n,m}^{r_i}$ be a random rank-$r_i$ matrix with i.i.d. entries in $\mathbb{R}^{m \times n}$. Then, with probability 1,*

$$\mathrm{rank}\left(\sum_{i=1}^{d} A_i\right) = \min\left(m, n, \sum_{i=1}^{d} r_i\right)$$

*If $\sum_{i=1}^{d} r_i \ge \min(n, m)$, $\sum_{i=1}^{d} A_i$ is a full-rank matrix with probability 1.*

*Proof.* Given the upper bound from Proposition E.6 and the Lemma F.6,

$$\mathrm{rank}\left(\sum_{i=1}^{d} A_i\right) \le \min\left(m, n, \sum_{i=1}^{d} r_i\right)$$

The lower bound derived in Proposition E.7 allows us to write,

$$\mathrm{rank}\left(\sum_{i=1}^{d} A_i\right) \ge \min\left(n, \sum_{i=1}^{d} r_i\right) + \min\left(m, \sum_{i=1}^{d} r_i\right) - \sum_{i=1}^{d} r_i$$

Let's set $r = \sum_{i=1}^{d} r_i$. If $r < \min(n, m)$ we have the result. Otherwise, let's assume $n \le r \le m$ which gives $n \ge \mathrm{rank}\left(\sum_{i=1}^{d} A_i\right) \ge n + r - r = n$ (the result is identical if we inverse the role of $n$ and $m$). Lastly, for $n \le m \le r$, this inequality boils down to $n + m - r \le \mathrm{rank}\left(\sum_{i=1}^{d} A_i\right) \le n$. This means that there is potentially $r - m$ columns that are linearly dependent of the others. However, by marginalization, each column $(\sum A_i)_{:j}$ of the sum is a random vector. As the columns of each $A_i$ are independent of all the others, the sum of the $j$-th columns of the $A_i$s are independent of the other columns of the sum. Thus, no columns of the sum is linearly dependent of the others so that $r = m$ and $\mathrm{rank}\left(\sum_{i=1}^{d} A_i\right) = n$ which concludes the proof. $\square$

## G  PROOF OF THEOREM 5.1

We restate the theorem of section 5.1.

**Theorem G.1.** *After $N \ge 0$ accumulation of the AccLoRT linear regression with rank $r$,*

$$\min_{A_N, B_N} \|Y - W_N X - A_N B_N X\|_F \le \sum_{i > (N+1)r} \sigma_i^2 + \varepsilon_{\mathrm{OLS}}$$

*where the $\sigma_i$'s correspond to the ordered singular values of $W_{\mathrm{OLS}} X = Y X^\top (X X^\top)^{-1} X$ and $\varepsilon_{\mathrm{OLS}} = \|Y - W_{\mathrm{OLS}} X\|_2$ is the non reachable error caused by the plausible non-linearity relation between $X$ and $Y$. Additionally, $W_{N+1} = U_{(N+1)r}^\top U_{(N+1)r} W_{\mathrm{OLS}}$ with $U_k$ the first $k$ left singular vectors of $W_{\mathrm{OLS}} X$.*

*Proof.* The proof will be tackled inductively. We will suppose that $n \leq m$ without loss of generality. First, we can simplify the problem by separating the non-linear relation as the sum of two norms

$$\min_{A_N, B_N} \|Y - W_{\text{OLS}}X\|_F + \|W_{\text{OLS}}X - W_N X - A_N B_N X\|$$

with the proper constraint on $W_N$, so that we will only concentrate on the last term.

For $N = 0$, the problem shrinks to $\min_{A,B} \|W_{\text{OLS}}X - ABX\|_F$ where the matrix $A$ and $B$ are of rank $r$ which is exactly the Reduced Rank Regression Izenman (1975). The solution is then given by $AB = U_r^\top U_r W_{\text{OLS}}$ where $U_r$ are the left singular vectors of $W_{\text{OLS}}X$. From there, $\|W_{\text{OLS}}X - ABX\|_F = \left\|(I - U_r^\top U_r)W_{\text{OLS}}X\right\|_F = \left\|(I - U_r^\top U_r)USV^\top\right\|_F$ and remarking that $(I - U_r^\top U_r)$ is the projector onto $n - r$ least singular vectors, we get that $\|W_{\text{OLS}}X - ABX\|_F = \sum_{i>r} \sigma_i^2$ from the definition of the Frobenius norm. We finish the base case by seeing that $W_1 = AB = U_r^\top U_r W_{\text{OLS}}$.

By induction, suppose the result is true for $N - 1$. Then, noting $\overline{W} = W_{\text{OLS}} - W_N$, we aim to find the solution to $\min_{A_N, B_N} \left\|\overline{W}X - A_N B_N X\right\|$. This is obviously another case of Reduced Rank Regression and reads $A_N B_N = \overline{U_r}^\top \overline{U_r W}$ as a solution, with $\overline{U}$ the left singular vectors of $\overline{W}X$. By the recurrence hypothesis, $W_N = U_{Nr}^\top U_{Nr} W_{\text{OLS}}$ so that $\overline{W} = (I - U_{Nr}^\top U_{Nr})W_{\text{OLS}}$. To conclude, we need to show that $W_N + A_N B_N = U_{(N+1)r}^\top U_{(N+1)r}$.

First, observe that $\overline{W} = W_{\text{OLS}} - W_N = (I - U_{Nr}^\top U_{Nr})W_{\text{OLS}} = U_{Nr+1:}^\top U_{Nr+1:} W_{\text{OLS}}$ where we use the slicing notation $U_{a:b}$ to indicate that we keep only the left singular vectors from the $a$-th to the $b - 1$-th vector.

$$
\begin{aligned}
W_{\text{OLS}}X - W_N X - A_N B_N X &= (I - U_{Nr}^\top U_{Nr})W_{\text{OLS}}X + \overline{U_r}^\top \overline{U_r W} X \\
&= (I - \overline{U_r}^\top \overline{U_r})(I - U_{Nr}^\top U_{Nr})W_{\text{OLS}}X \\
&= \underbrace{\overline{U_{r+1:}}^\top \overline{U_{r+1:}}}_{\text{discard the } r \text{ first l.s vectors of } \overline{W}X} \underbrace{U_{Nr+1:}^\top U_{Nr+1:} W_{\text{OLS}}X}_{\text{discard the first } Nr \text{ l.s vectors of } W_{\text{OLS}}X} \\
&= U_{Nr+1:(N+1)r}^\top U_{Nr+1:(N+1)r} U_{Nr+1:}^\top U_{Nr+1:} W_{\text{OLS}}X \\
&= U_{(N+1)r+1:}^\top U_{(N+1)r+1:} W_{\text{OLS}}X
\end{aligned}
$$

Thus,

$$
\begin{aligned}
\|W_{\text{OLS}}X - W_N X - A_N B_N X\|_F &= \left\|U_{(N+1)r+1:}^\top U_{(N+1)r+1:} W_{\text{OLS}}X\right\|_F \\
&= \sum_{i>(N+1)r} \sigma_i^2
\end{aligned}
$$

with $\sigma_i$'s the singular values of $W_{\text{OLS}}X$ and from the previous derivation $W_{N+1} = W_N + A_N B_N = U_{(N+1)r+1:}^\top U_{(N+1)r+1:} W_{\text{OLS}} - W_{\text{OLS}} = U_{(N+1)r}^\top U_{(N+1)r} W_{\text{OLS}}$. □

# H    On the SGD updates of AccLoRT

This section will be reserved to the study of how the rank of a AccLoRT layer fluctuates during the training. Recall that a AccLoRT layer is given by $y = (W + AB)x$ where $y \in \mathbb{R}^n, W \in \mathbb{R}^{n \times m}, x \in \mathbb{R}^m$ and $A \in \mathbb{R}^{n \times r}, B \in \mathbb{R}^{r \times m}$. The gradient of the matrices $A, B$ are then

$$\frac{\partial \mathcal{L}}{\partial A} = \left(\frac{\partial \mathcal{L}}{\partial y} x^\top\right) B^\top \qquad \frac{\partial \mathcal{L}}{\partial B} = A^\top \left(\frac{\partial \mathcal{L}}{\partial y} x^\top\right)$$

Under Gradient Descent, the optimizer updates are $A := A - \eta \frac{\partial \mathcal{L}}{\partial A}, B := B - \eta \frac{\partial \mathcal{L}}{\partial B}$. A immediate question that arises is how does the ranks of $A, B$ increase after one update. The gradient $\mathcal{L}$ with respect to both $A$ and $B$ is the product of a matrix and an outer product of two vectors (the previous gradient and the input) in the case of (stochastic) gradient descent as we consider batchsize of 1.

First, we recall a classical lemma (as for $v \in \text{Im}(AB), v = ABu = A(Bu)$ for some vector $u$),

**Lemma H.1.** *Let $A \in \mathbb{R}^{n \times r}, B \in \mathbb{R}^{r \times m}$, $\mathrm{col}(AB) \subset \mathrm{col}(A)$ and $\mathrm{row}(AB) \subset \mathrm{row}(B)$.*

**Lemma H.2.** *Let $A \in \mathbb{R}^{n \times r}, u \in \mathbb{R}^r, v \in \mathbb{R}^m$ with $u \notin \mathrm{Ker}(A), v \neq 0$, then*

$$\mathrm{rank}(Auv^\top) = 1 \qquad \mathrm{col}(Auv^\top) = \mathrm{span}(\{Au\}) \qquad \mathrm{row}(Auv^\top) = \mathrm{span}(\{v\})$$

*Proof.* First, for $u, v \neq 0$, $\mathrm{rank}(uv^\top) = 1$. The column of $uv^\top$ are given by $(v_1 u, \ldots, v_j u)$ hence, $\mathrm{Im}(uv^\top) = \mathrm{col}(uv^\top) = \mathrm{span}(\{u\})$ and by transposing, $\mathrm{row}(uv^\top) = \mathrm{span}(\{v\})$.

Now, to conclude the proof, simply observe that $Auv^\top = (Au)v^\top$ and that $Au$ is a vector in $\mathbb{R}^n$ thus $Auv^\top$ is an outer product and has rank 1 if $Au \neq 0$. $\qquad \square$

Using the previous lemma and the fact that $\mathrm{col}(A^\top) = \mathrm{row}(A), \mathrm{col}(A) = \mathrm{row}(A^\top)$, we have that

$$\mathrm{col}\left(\frac{\partial \mathcal{L}}{\partial A}\right) = \mathrm{span}\left(\left\{\frac{\partial \mathcal{L}}{\partial y}\right\}\right) \qquad\qquad \mathrm{row}\left(\frac{\partial \mathcal{L}}{\partial A}\right) = \mathrm{span}(\{Bx\})$$

$$\mathrm{col}\left(\frac{\partial \mathcal{L}}{\partial B}\right) = \mathrm{span}\left(\left\{A^\top \frac{\partial \mathcal{L}}{\partial y}\right\}\right) \qquad\qquad \mathrm{row}\left(\frac{\partial \mathcal{L}}{\partial B}\right) = \mathrm{span}(\{x\})$$

Hence, from the Proposition E.4, we have the lower bound,

$$\mathrm{rank}\left(A - \eta \frac{\partial \mathcal{L}}{\partial A}\right) \geq \mathrm{rank}(A) + 1 - \dim\left(R_A \cap R_{\frac{\partial \mathcal{L}}{\partial A}}\right) - \dim\left(C_A \cap C_{\frac{\partial \mathcal{L}}{\partial A}}\right)$$

$$\geq \mathrm{rank}(A) + 1$$
$$- \dim\left(\mathrm{row}(A) \cap \mathrm{span}(\{Bx\})\right)$$
$$- \dim\left(\mathrm{col}(A) \cap \mathrm{span}\left(\left\{\frac{\partial \mathcal{L}}{\partial y}\right\}\right)\right)$$

A simple bound for the update of the $A$ matrix is that

$$\mathrm{rank}(A^{t+1}) \in \{\mathrm{rank}(A^t) - 1, \mathrm{rank}(A^t), \mathrm{rank}(A^t) + 1\}$$

In an equivalent manner, a lower bound for $B_i$ can be found as

$$\mathrm{rank}\left(B - \eta \frac{\partial \mathcal{L}}{\partial B}\right) \geq \mathrm{rank}(B) + 1 - \dim\left(R_B \cap R_{\frac{\partial \mathcal{L}}{\partial B}}\right) - \dim\left(C_A \cap C_{\frac{\partial \mathcal{L}}{\partial A}}\right)$$

$$\geq \mathrm{rank}(B) + 1$$
$$- \dim\left(\mathrm{row}(B) \cap \mathrm{span}(\{x\})\right)$$
$$- \dim\left(\mathrm{col}(B) \cap \mathrm{span}\left(\left\{A^\top \frac{\partial \mathcal{L}}{\partial y}\right\}\right)\right)$$

So that,

$$\mathrm{rank}(B^{t+1}) \in \{\mathrm{rank}(B^t) - 1, \mathrm{rank}(B^t), \mathrm{rank}(B^t) + 1\}$$

These bounds confirm solely our intuition that substracting a matrix by a rank one matrix can only either kill a dimension (imagine a matrix with $n$ columns equal to the i-th column of $A$), do nothing if the rank-1 matrix can be formed by linear combination of columns of $A$ or increase the rank by 1 if it is generated by a free vector from the columns of $A$. However, we can find the exact dimension of these intersections of space which result in the following theorem.

**Theorem H.3.** *Consider any neural network under stochastic gradient descent optimization with a LoRA layer $y = Wx + ABx$ with $W \in \mathbb{R}^{n \times m}$ frozen and a rank $r < \min(n, m)$. Suppose that the training dataset has no duplicates and that the initial matrices are $A_0 = 0$ and $B_0$ has elements drawn from a distribution absolutely continuous with respect to the Lebesgue measure.*
*Then, with probability one, for a batchsize $d$, at time $t$,*

$$\mathrm{rank}(A_t) = \min(d \times t, r) \qquad \mathrm{rank}(B_t) = r$$

*Proof.* The proof will be structured in four parts, an introduction to some notations, expressing $B_t$ as a function of $B_0$ and the latter two parts relying on the second result.

**Notations & Dataset consideration** Following the beginning of the section, we consider $x$ to be the input of the LoRA layer and $y$ its output such that $y = (W + AB)x$. We also note $z$ to be the tokenized vectors in $V^d$ ($d$ being the maximum sequence length) and $f$ a function of the previous layers such that $x = f(z)$. When a subscript is added to any $A, B, x, y$ or $z$ it specifies which training steps we examine except when it is obvious like $I_m$ the identity matrix of shape $(m, m)$.

Although the dataset is not absolutely continuous with respect to the Lebesgue measure (which would considerably simplify the results), in pretraining large language models, we usually make use of extremely large dataset like the C4 dataset Raffel et al. (2023) were no repetition occurs and the architecture we train have a significant vocabulary $V$ (32000 for Llama models) with large feature dimension $m$ and a maximum sequence length $d$ greater or equal than 256. This means that the dataset is discrete and can be represented from a fixed embedding matrix as a look-up matrix probability of shape $|V|^d \times dm$ as there would be $|V|^d$ possible sentences that can be formed which would be mapped to a vector in $\mathbb{R}^{dm}$. As the embedding matrix is full-rank by random consideration (generalized randomly and the backpropagation does not create any dependence), it can be shown that the probability matrix is full rank for $|V| > 1$ as $|V|^d > dm$ for $m < |V|^d/d$ which is almost always the case for LLMs (otherwise the rank is $(|V| - 1)d + 1$ but we will not prove it here). This means that any batch of size lesser than $dm$ would form a linearly independent family.

**Expression of $B_t$ as $B_0 C_t$** For $t = 0$, $B_t = B_0 I_m$ and $C_0 = I_m$. By induction on $t \in \mathbb{N}$, suppose that for all integers $s \le t$, there exists $C_s \in \mathbb{R}^{m \times m}$ such that $B_s = B_0 C_s$. Let $J_s = \frac{\partial \mathcal{L}}{\partial y_s} x_s^\top$ for any $s \in \mathbb{N}$. Then

$$
\begin{aligned}
B_{t+1} &= B_0 C_t - \eta A_t^\top J_t \\
&= B_0 C_t + \eta^2 B_0 C_{t-1} J_{t-1} J_t - \eta A_{t-1}^\top J_t \\
&= B_0 \left( C_t + \eta^2 C_{t-1} J_{t-1} J_t + \eta^2 C_{t-2} J_{t-2} J_t \right) - \eta A_{t-2}^\top J_t \\
&= B_0 \left( C_t + \eta^2 \sum_{k=1}^{t} C_{t-k} J_{t-k} J_t \right) \\
&= B_0 C_{t+1}
\end{aligned}
$$

as $A_0 = 0$. This concludes the first part of the proof with $C_{t+1} = C_t + \eta^2 \sum_{k=1}^{t} C_{t-k} J_{t-k} J_t$.

**Constant rank for $B_t$** To show that $B$ has constant rank, we will show show that $\dim\left( \text{row}(B) \cap \text{span}\left( \{x\} \right) \right) = 0$ and $\dim\left( \text{col}(B) \cap \text{span}\left( \left\{ A_i^\top \frac{\partial \mathcal{L}}{\partial y} \right\} \right) \right) = 1$. This will result in the lower bound $\text{rank}(B_{t+1}) \ge \text{rank}(B_t)$. By induction, knowing that $B_0$ is full rank, $B_t$ will be full rank for any step $t$.

By induction, suppose that $B_t$ is full rank with probability one. From Proposition F.3, the family $\{b_1, \ldots, b_r, x\}$ is linearly independent with probability 1 where $b_i$ is the $i$-th row of $B_t$. This is true because on the one hand the rows of $B_t$ form an already linearly independent family and $x$ is drawn from a discrete set of size $|V|^d$ as discussed previously. Thus,

$$
P\left( \dim\left( \text{row}(B_t) \cap \text{span}\left( \{x\} \right) \right) = 1 \right) = P(\{b_1, \ldots, b_r, x\} \text{ is linearly dependent}) = 0
$$

Then, with probability 1, $\dim\left( \text{row}(B_t) \cap \text{span}\left( \{x\} \right) \right) = 0$.

Now, for the second dimension, as $B_t$ is full rank by the induction hypothesis, if $A_t^\top$ is a function of the form $B_t C$ for some $C$ matrix (Lemma H.1), it will conclude the argument because its image would then lie in the image of $B_t$ meaning that the intersection of both spaces is not empty and thus of dimension 1. To prove such result, see that $B_t$ is of the form $B_0 C_t$ from the beginning of the proof. As both are full rank, $\text{col}(B_t) = \text{col}(B_0)$ and the previous idea should apply similarly if $A_t^\top$ is function of $B_0$. From the same proposition, the vector $A_t^\top \frac{\partial \mathcal{L}}{\partial y}$ has the adequate form. There is one last thing to verify: that $\frac{\partial \mathcal{L}}{\partial y}$ is not in the kernel of $A_t^\top$. If it is the case, this would diminish the rank of $B_{t+1}$.

But, recognizing that the space $\text{Ker}(A_t^\top)$ is a subspace of $\mathbb{R}^n$ of dimension $< n$, it is thus of zero measure for the Lebesgue measure (see the end of the proof of Proposition F.3). The probability that

the gradient of the loss function with respect to the output of the LoRA layer is in the kernel of $A_t$ is equal to 0.

**Rank increase of $A_t$**   A similar argument from $B$ case shows that $\dim\left(\mathrm{col}(A_t) \cap \mathrm{span}\left(\left\{\frac{\partial \mathcal{L}}{\partial y}\right\}\right)\right) = 0$ with probability 1. It can be shown that while $\mathrm{rank}(A) < r$, the intersection is empty ($= \{0\}$). Indeed, $B_t x = B_0 C_t x$ lies in the image of $B_0$ by Lemma H.1. Let $a_1, \ldots, a_r$ be a basis of the rows of $A_t$, consider the family $\{a_1, \ldots, a_r, B_t x\}$. By the Proposition F.3, the probability that this family is linearly dependent is equal to zero. Hence, $B_t x$ cannot be written as a linear combination of the $a_i$s and the space it spanned does not intersect the rows of $A$ (except for the zero vector). Thus, $\mathrm{col}(A_t) \cap \mathrm{span}\left(\left\{\frac{\partial \mathcal{L}}{\partial y}\right\}\right) = \emptyset$ and this finalizes the proof.

**Extension to any batchsize**   The proof is easily adapted for a batchsize $d$ greater than 1 as one would replace $x$ by a collection of $x_1, \ldots, x_d$, the gradients of the loss with respect to $A$ and $B$ would then consists of rank $d$ matrices instad of 1 and from the discussion of the beginning of the proof, the result follows. □

**Corollary H.4.** *Under the same assumptions as 3.3, replacing the LoRA layer with a AccLoRT layer with accumulation each $T$ step yields at time $t$,*

$$\mathrm{rank}(A_t) = \min(t \bmod T, r) \quad \mathrm{rank}(B_t) = r$$

*Proof.* This is a simple application of the previous theorem, in which the matrix $A$ and $B$ are reset introducing a modulo on the accumulation frequency. □

