# OpenReview forum: "AccLoRT: Efficient Large Language Models Pretraining through Low-Rank Accumulation"
_ICLR.cc/2026/Conference — Submitted to ICLR 2026_

### Official Review · Reviewer_13sN · 2025-10-25

**Soundness:** 3
**Presentation:** 2
**Contribution:** 2
**Rating:** 4
**Confidence:** 5

**Summary:**

This paper proposes AccLoRT (Accumulated Low-Rank Training), a memory-efficient pretraining framework for large language models (LLMs). The method sequentially trains multiple low-rank matrices and accumulates them into a frozen high-rank matrix, enabling fully low-rank training throughout the entire pretraining process without any full-parameter warm-up. The authors provide theoretical analyses on the rank bounds of matrix sums, LoRA’s rank evolution under SGD, and the asymptotic equivalence between AccLoRT and full-rank training. Extensive experiments on Llama models (60M–1B) demonstrate that AccLoRT achieves superior perplexity and memory efficiency compared to existing methods such as GaLore, ReLoRA, and SLTrain.

**Strengths:**

1. The paper  derives upper/lower rank bounds and theorems describing LoRA’s rank evolution and AccLoRT’s convergence, giving a clear mathematical understanding of why accumulation works.

2. Empirical evaluations span a wide range of model sizes (60M–1B) shows improvements.

3. The proposed idea is good and reasonable.

**Weaknesses:**

1. The paper completely omits "Fira: Can We Achieve Full-rank Training of LLMs Under Low-rank Constraint?"[1], a closely related approach that also performs full-rank pretraining under low-rank constraints. Both share nearly identical experimental settings (rank, optimizer, and dataset) and similar motivation (achieving full-rank training). Without detailed discussion in motivation, methodology, memory consumption, and experimental performance comparison, the claimed advantages of AccLoRT remain incomplete and potentially overstated.

2. Current model size in experiments is small. It would be better to see pretrain experiments on 7B models. In this experimental setting, both Fira [1], Galore [2],  APOLLO [3] all conduct the 7B experiments.

3. Illustration is poor. For example, the font size in Figure 1 and Figure 3 is too small to read for potential readers.


[1] Fira: Can We Achieve Full-rank Training of LLMs Under Low-rank Constraint?

[2] GaLore: Memory-Efficient LLM Training by Gradient Low-Rank Projection

[3] APOLLO: SGD-like Memory, AdamW-level Performance

**Questions:**

see weaknesses

I will adjust my score according to the rebuttal.

---

> ### Author Response · Authors · 2025-11-22
> **Response to Reviewer 13sN**
>
> Dear Reviewer 13sN, we appreciate your insightful comments. In the following response, we address each of your concerns point-by-point and have updated the paper to reflect these changes. We welcome any further questions you may have.
>
> ---
>
> > **(Weakness 1)** *The paper completely omits "Fira: Can We Achieve Full-rank Training of LLMs Under Low-rank Constraint?"[1], a closely related approach that also performs full-rank pretraining under low-rank constraints. Both share nearly identical experimental settings (rank, optimizer, and dataset) and similar motivation (achieving full-rank training). Without detailed discussion in motivation, methodology, memory consumption, and experimental performance comparison, the claimed advantages of AccLoRT remain incomplete and potentially overstated.*
>
>
> **Re:**
> We sincerely apologize for the oversight of Fira [1] in our initial submission. We recognize it as a closely related approach and note its recent acceptance to NeurIPS 2025. We have now incorporated a detailed discussion in the revised Related Works, acknowledging its substantial contribution.
>
> While both methods share the goal of full-rank training under low-rank constraints, we discussed their distinction in the following:
>
> **Methodology**: Fira primarily focuses on gradient or update decomposition. In contrast, AccLoRT introduces a sequential subspace learning paradigm, systematically constructing the parameter space by accumulating subspaces during training.
>
> **Theory**: AccLoRT distinguishes itself with a rigorous rank evolution theory (Theorem 3.3) and a pareto frontier analysis. This provides a theoretical framework for understanding when low-rank accumulation achieves full-rank coverage, offering principled guidance for hyperparameter selection
>
>
> ---
>
> > **(Weakness 2)** *Current model size in experiments is small. It would be better to see pretrain experiments on 7B models. In this experimental setting, both Fira [1], Galore [2], APOLLO [3] all conduct the 7B experiments.*
>
> *[1] Fira: Can We Achieve Full-rank Training of LLMs Under Low-rank Constraint?*
>
> *[2] GaLore: Memory-Efficient LLM Training by Gradient Low-Rank Projection*
>
> *[3] APOLLO: SGD-like Memory, AdamW-level Performance*
>
>
> **Re**: We appreciate the reviewers's constructive comment on the experiments of 7B's model. We have presented the results of AccLoRT, please see Figure 8 of revised paper for more details.
>
>
>
> ---
>
> > **(Weakness 3)** *Illustration is poor. For example, the font size in Figure 1 and Figure 3 is too small to read for potential readers.*
>
> **Re**: We thank the reviewer for the feedback regarding the readability of our illustrations. We have revised Figures 1 and 3 by increasing the font sizes to ensure they are clear for readers, please refer to the revised paper.

---

> ### Comment · Reviewer_13sN · 2025-11-22
>
> Thanks for the rebuttal. My concerns are partly addressed. Therefore, I increase my score to borderline accept.
>
> However, I think only putting the results of AccLoRT in Figure 8 is **meaningless without any comparison** with previous methods. I'd like to see the comparison of the proposed method with Fira at 7B scale to fully valid its effectiveness.

---

> > ### Author Response · Authors · 2025-11-26
> > **Response to Reviewer 13sN**
> >
> > We sincerely thank the reviewer for the score increase and the suggestion regarding the 7B comparison. We are actively conducting this experiment and will update the paper once we have the results.
> >
> > We would like to share our current progress:
> >
> > - Our standard clusters are equipped with A100 40GB GPUs. On this hardware, AccLoRT runs efficiently with a micro_batch=4. However, when setting up the baselines, we observed that Fira and GaLore require approximately 42GB+ of memory (even with micro_batch=1) for the 7B model. Consequently, we have to conduct the 7B experiments on RTX A6000 48GB servers, which substantially extends the training time.
> >
> > - Given the discussion deadline (Dec 2), we are prioritizing the comparison. While the full training cycle might not be completed by the deadline, we will update the paper with the latest available results to demonstrate the effectiveness.
> >
> > We will post a follow-up comment as soon as the plot is updated.

---

### Official Review · Reviewer_Et4y · 2025-10-26

**Soundness:** 2
**Presentation:** 2
**Contribution:** 2
**Rating:** 4
**Confidence:** 3

**Summary:**

The paper proposes a method to pretrain model by accumulating low rank updates. In the theory part, the paper provides upper and lower bounds for the weight matrix rank after accumulating the low rank updates and bounds for the ranks of adapter matrices $A,B$ with gradient updates. In experiment, the paper proposes to separate between the first initialization and subsequent ones after each merge, where in the first initialization $A,B$ are obtained by the truncated QR decomposition of a normal matrix and in the subsequent initializations, $A$ is 0 and $B$ is normal. The experiment shows that the proposed method has better perplexity on the pretrain task using Llama, and is competitive with other methods on fine-tune tasks.

**Strengths:**

- There have been several works on using accumulating LoRA, but understanding the rank evolvement during training remains not well studied. This paper provides both upper and lower bounds for this question, which I appreciate.
- The proposed method is simpler than similar existing approaches, such as ReLoRA. In particular, ReLoRA requires a warm up stage for the full model while this method doesn't.
- Empirically, the method performs better than existing approaches with memory footprint.

**Weaknesses:**

- First of all, the paper lacks clarity in the representation.I'm not sure I really understand the proposed method.
  - In particular, how is the full model $W$ initialized? Is it to 0? If this is the case, it very surprising to me.
  - Theorem 3.3 also doesn't state the necessary assumptions, such as the initialization of $A_0$, $B_0$. Its proof in the appendix is also not clearer. For example, line 1304 says the input $x$ is drawn from a discrete set referred in line 1274 - 1284 where some assumptions about this set is made, which I'm also not sure I understand.
- The algorithmic contribution of this paper to me is quite incremental. It boils down to finding a different initialization of $A,B$. The idea of merging the adapters to the base model has been explored quite a lot before.
- To this end, I'm not sure I understand where the improvements in the experiment comes from. In particular, compared with ReLoRA which basically uses the same merging idea, and even warms up the model before the low-rank update phase, why does the proposed method perform better?

**Questions:**

- Please see the weaknesses.
- In table 2, can the authors provide the comparison with the other methods in Table 3?
- Line 215 mentioned $W_{acc}$, Is there a difference with the weight $W$ in Algorithm 1?
- In Algorithm 1, what is the goal of the if-else statement in Line 224-227? It doesn't seem the be explained in the text.

---

> ### Author Response · Authors · 2025-11-22
> **Response to Reviewer Et4y (1/3)**
>
> Dear Reviewer Et4y, we appreciate your insightful comments. In the following response, we address each of your concerns point-by-point and have updated the paper to reflect these changes. We welcome any further questions you may have.
>
> ---
>
> > **(Weakness 1)**: *First of all, the paper lacks clarity in the representation.I'm not sure I really understand the proposed method.*
>
> > *In particular, how is the full model initialized? Is it to 0? If this is the case, it very surprising to me.*
>
> > *Theorem 3.3 also doesn't state the necessary assumptions, such as the initialization of $A_0$, $B_0$. Its proof in the appendix is also not clearer. For example, line 1304 says the input  is drawn from a discrete set referred in line 1274 - 1284 where some assumptions about this set is made, which I'm also not sure I understand.*
>
>
> **Re**: **The model is not initialized to zero**. To clarify, at the very beginning of training ($t=0$), the model is not initialized to zero. Instead, we employ a truncated-QR initialization strategy. We first sample a target weight matrix $W_0$ from a standard distribution (e.g., Kaiming or Gaussian) and then perform a truncated QR decomposition to obtain $[B, A] = \text{QR}(W_0, r)$. This ensures that the product $BA$ approximates the standard initialization of a full-rank model, which we found to be significantly more effective than initializing low-rank matrices $A$ and $B$ independently (as shown in Figure 3).
>
>
> **More details for Theorem 3.3.** Thank you for pointing out this oversight. In Theorem 3.3, we assume the initialization scheme introduced in the LoRA paper: specifically, $A_0$ is initialized to zero, while $B_0$ is drawn from a Kaiming-uniform or Gaussian distribution. Under this setting, the resulting matrix $B_0$ (of size $r \times m$) is full-rank almost surely. More generally, this assumption holds for any $B_0$ initialized from a distribution that is absolutely continuous with respect to the Lebesgue measure. Indeed, the proof fundamentally relies on the premise that the initial matrix is full-rank. This property is formally derived from the following theorem:
>
> **Theorem** (see [1]) A polynomial function on $\mathbb{R}^n$ to $\mathbb{R}$, is either identically 0, or non-zero almost everywhere (for the Lebesgue measure).
>
> A direct corollary is that the set of singular matrices is negligible in $\mathbb{R}^{n \times n}$ under any measure absolutely continuous with respect to the Lebesgue measure. This holds because the determinant is a polynomial function, and its zero set (the singular matrices) constitutes a null set. Thus, any matrix with random coefficients drawn from such distributions is guaranteed to be full-rank almost surely.
>
>
> **Regarding the dataset assumptions**. We rely on the premise that in high-dimensional spaces (with large vocabulary and embedding dimensions), the probability of encountering linearly dependent inputs is negligible. Specifically, we distinguish between the model input (token sequences) and the AccLoRT layer input (latent representations output by the preceding layer).For the theorem to hold, we assume that the inputs within a mini-batch are unique. Given the non-linear transformations of the initial layers, distinct token sequences in a batch are mapped to linearly independent hidden states almost surely, provided the batch size is small relative to the hidden dimension ($b \ll d$). This linear independence is crucial for the proof regarding the rank increase of $A$: it ensures that the set of vectors $\{a_1, \dots, a_r, B_t x_1, \dots, B_t x_d\}$ remains linearly independent, thereby validating the rank expansion stated in the theorem.
>
>
> [1] Richard Caron \& Tim Traynor, _The zero set of a polynomial_

---

> ### Author Response · Authors · 2025-11-22
> **Response to Reviewer Et4y (2/3)**
>
> > **(Weakness 2)** *The algorithmic contribution of this paper to me is quite incremental. It boils down to finding a different initialization of $A_0$,  $B_0$. The idea of merging the adapters to the base model has been explored quite a lot before.*
>
>
> **Re**: We appreciate the reviewer’s constructive feedback. We acknowledge that AccLoRT appears to share conceptual similarities with prior "merging" approaches like ReLoRA. However, we would like to clarify that the specific initialization of low-rank matrices is not the central contribution of our work.
>
> Instead, our core contribution lies in breaking the prevailing assumption that a full-parameter warmup is a prerequisite for effective low-rank pretraining, thereby establishing a novel training paradigm.
>
>
> We distinguish our contribution from incremental improvements in three key aspects:
>
> **1) Removing the Warmup Dependency.** Previous state-of-the-art methods (e.g., ReLoRA) operate on the belief that low-rank training is insufficient to capture the model's initial optimization trajectory, thus necessitating a high-cost full-parameter warmup. AccLoRT fundamentally challenges this view. By introducing sequential low-rank training, we demonstrate that it is possible to train a model from scratch using strictly low-rank updates without any full-rank warmup. This ensures that the entire training pipeline remains strictly memory-efficient from start to finish.
>
> **2）Sequential Subspace Learning Paradigm.** AccLoRT systematically builds weights from scratch by sequentially accumulating low-rank subspaces. We demonstrate that this sequential subspace learning is inherently more effective than full-parameter warmup, as it forces the model to incrementally capture informative structures, ensuring stable and fast convergence.
>
>
> **3) Theoretical Framework for Rank Evolution.** Beyond empirical results, we provide a rigorous theoretical characterization of how rank accumulates and when full-rank coverage is achieved (Theorem 3.3). This transforms a heuristic trick into a principled strategy, offering guidelines that are absent in previous works.
>
> We hope this clarification helps disentangle our improvement on previously existing methods and highlights the conceptual and theoretical contributions that extend beyond an initialization change.
>
> ---
>
>
> > **(Weakness 3)** *To this end, I'm not sure I understand where the improvements in the experiment comes from. In particular, compared with ReLoRA which basically uses the same merging idea, and even warms up the model before the low-rank update phase, why does the proposed method perform better?*
>
> **Re:** We appreciate the reviewer's question regarding the source of improvement. The key difference lies in the fundamental learning dynamics of the two methods:
>
> **1. ReLoRA: Random Refinement on a Fixed Basis.** Because of the full-parameter warmup, ReLoRA essentially operates as a restarted fine-tuning method. By the time the low-rank phase begins, the model is already full-rank. Consequently, the low-rank adapters simply refine the weights in random low-rank directions dictated by the current batch gradients.
>
> **2. AccLoRT: Subspace Filling.** In contrast, AccLoRT explicitly constructs the parameter space by filling it with a sequence of orthogonal subspaces. Starting from a low-rank initialization, each accumulation step introduces a new, orthogonal subspace of dimension $r$. The model is forced to optimize within this specific subspace before moving to the next. This creates a "sequential pretraining" dynamic: instead of randomly refining a pre-warmed model, AccLoRT systematically builds the full-rank matrix component by component.
>
> **Why this leads to improvement:** This sequential subspace filling ensures that the model prioritizes learning the most informative subspaces first (as guided by the optimization landscape). This dynamic is fundamentally different from ReLoRA's random refinements. While theoretical proofs for LLMs are challenging, our empirical results on linear regression (Section 5.1) clearly validate this subspace filling hypothesis, showing that AccLoRT effectively recovers the target weights through this sequential process.

---

> ### Author Response · Authors · 2025-11-22
> **Response to Reviewer Et4y (3/3)**
>
> > **(Question 1)** *In table 2, can the authors provide the comparison with the other methods in Table 3?*
>
> **Re**: Yes, we have provided the comparison with other methods in Table 11, please refer to Appendix for more details.
>
> ---
>
> > **(Question 2)** *Line 215 mentioned W_acc, Is there a difference with the weight W in Algorithm 1?*
>
> **Re**: Yes, it is the same matrix in Algorithm 1. We have revised Algorithm 1 now.
>
> ---
>
> > **(Question 3)** *In Algorithm 1, what is the goal of the if-else statement in Line 224-227? It doesn't seem the be explained in the text.*
>
> **Re**: We apologize for the omission. Lines 224-227 implement a dynamic storage strategy designed to optimize memory usage by switching between factorized and merged storage modes. In the early training stages (the if condition), maintaining updates in their factorized form consumes significantly less memory than materializing the full dense matrix. Crucially, this memory enables the adaptive rank strategy detailed in Appendix B.5: the saved memory allows us to temporarily increase the rank of AccLoRT in the early stages (as defined in Eq. 2 of Appendix) to accelerate convergence without exceeding the total memory budget.

---

### Official Review · Reviewer_UsMi · 2025-10-27

**Soundness:** 3
**Presentation:** 3
**Contribution:** 3
**Rating:** 6
**Confidence:** 3

**Summary:**

This paper introduces AccLoRT, a framework for memory-efficient pretraining of large language models (LLMs) based on sequential accumulation of low-rank matrices. The core idea is to sidestep the memory bottleneck of full-rank training by progressively training and freezing low-rank adapters. Theoretical analyses are provided, establishing upper and lower bounds for the rank of summed matrices, and elucidating rank dynamics during training. Extensive experiments on synthetic regression and Llama models from 60M to 1B parameters are presented, showing empirical benefits in memory usage and perplexity relative to prior methods. The approach is evaluated both in pretraining and fine-tuning scenarios.

**Strengths:**

1. The memory efficiency challenge in LLM pretraining is timely and relevant. The authors provide a compelling motivation, emphasizing the increasing computational inaccessibility of fundamental LLM research.
2. AccLoRT’s approach to accumulating low-rank matrices is carefully described, including specific details on initialization, memory trade-offs, and update mechanisms.
3. The experiments cover both toy and large-scale settings. Figure 3 reveals the effect of initialization on loss and perplexity for Llama models, giving actionable insight into implementation choices.
4. Table 3 demonstrates AccLoRT achieving strong or comparable perplexity to full-rank and other efficient training methods (like GaLore, ReLoRA, SLTrain) across various Llama model sizes—often with meaningful memory savings (as detailed in Table 2).

**Weaknesses:**

1. While the related work section summarizes many recent memory-efficient fine-tuning and pretraining approaches, it overlooks several directly relevant recent advances in low-rank and model compression for LLMs.
2. While Table 2 comprehensively compares model sizes and memory/parameter usage, Table 3's comparison is focused on perplexity only.
3. The framing and experiments focus exclusively on LLMs and linear regression; generalizability to vision models, multi-modal LLMs, or other architectures is not tested, even though low-rank techniques are often portable.

**Questions:**

1. Table 3 shows that AccLoRT marginally underperforms GaLore/full-rank at the 1B parameter scale, but outperforms on smaller models; do the authors attribute this to the hyperparameters, the method itself, or intrinsic limitations of low-rank accumulation as dimensionality increases?
2. Is there an avenue for automating the choice of rank $r$ and accumulation frequency $T$ (possibly dynamically during training) as opposed to hand-tuning or grid search?
3. Could the memory savings be further quantified in terms of wall-clock time, power consumption, or batch size enabled, especially on single-GPU or low-resource devices (beyond the current synthetic/LLama runs)?

---

> ### Author Response · Authors · 2025-11-22
> **Response to Reviewer UsMi**
>
> Dear Reviewer UsMi, we appreciate your insightful comments. In the following response, we address each of your concerns point-by-point and have updated the paper to reflect these changes. We welcome any further questions you may have.
>
>
> ---
>
> > **(Weakness 1)** *While the related work section summarizes many recent memory-efficient fine-tuning and pretraining approaches, it overlooks several directly relevant recent advances in low-rank and model compression for LLMs.*
>
> **Re**: We have added more recent related low-rank decomposition-based model compressions methods in related works, please refer to the revised related works for more details.</span>
>
>
> ---
> > **(Weakness 2)** *While Table 2 comprehensively compares model sizes and memory/parameter usage, Table 3's comparison is focused on perplexity only.*
>
> **Re**:  Yes, Table 2 compares the model sizes and memory usage for methods that used in Table 3. Since these two tables are corresponding to one experiment, we have added more clarity on the caption of Table 3 for their connections.</span>
>
> ---
> > **(Weakness 3)** *The framing and experiments focus exclusively on LLMs and linear regression; generalizability to vision models, multi-modal LLMs, or other architectures is not tested, even though low-rank techniques are often portable.*
>
> **Re**: We prioritized LLMs to align with standard memory-efficient benchmarks (e.g., GaLore, ReLoRA). However, AccLoRT is inherently architecture-agnostic. Its core mechanism, which performs fully low-rank training to approximate full-rank weights, relies on the spectral properties of linear layers. These are mathematical principles independent of data modality. Since modern vision (e.g., ViT) and multi-modal models share the same Transformer backbone, AccLoRT can be directly transferable to these architectures without modification. We have also added this discussions in the Conclusion section of the revised paper.
>
> ---
> > **(Question 1)** *Table 3 shows that AccLoRT marginally underperforms GaLore/full-rank at the 1B parameter scale, but outperforms on smaller models; do the authors attribute this to the hyperparameters, the method itself, or intrinsic limitations of low-rank accumulation as dimensionality increases?*
>
> **Re**: We applogize for the typo in the preliminary submission. AccLoRT achieves 15.49 in Table 3 on 1B model and outperforms Galore.
> In our main text, we mistakenlly stated that AccLoRT achieves 16.61. We have revised the main text now.
>
> ---
> > **(Question 2)** *Is there an avenue for automating the choice of rank and accumulation frequency (possibly dynamically during training) as opposed to hand-tuning or grid search?*
>
> **Re**:
> Yes, we provide a principled guideline for selecting the accumulation frequency based on the rank. Typically, the rank is determined by the available memory budget. The accumulation frequency is then set according to the proportion of training steps required to achieve full-rank coverage. Specifically, we define it as:
> $$
> T = \frac{t_{\text{total}}}{n/r} \times C,
> $$
> where $t_{\text{total}}$ is the number of full-parameter training steps, $n$ is the average matrix size, $r$ is the low-rank value, and $C \in [0.6, 1]$ is a scaling constant depending on the target convergence behavior. In the newly added Figure 9 and Figure 10, we investigate the stability of the above strategy, which shows that giving $r=100$, the frequency should be [1000, 2000] and giving $r=128$, the frequency should be [1500, 2500]. In Figure 9 and Figure 10, we show that we can obtain very stable performance when using the criterio.
> This setting has also shown to be **effective across various models** in our experiments. See Appendix B.2 for more details.
>
> ---
> > **(Question 3)** *Could the memory savings be further quantified in terms of wall-clock time, power consumption, or batch size enabled, especially on single-GPU or low-resource devices (beyond the current synthetic/LLama runs)?*
>
> **Re**: We appreciate the reviewer's suggestion to quantify practical efficiency. We would like to clarify that on the single-GPU/low-resource devices where AccLoRT is most beneficial, full-parameter training typically triggers Out-Of-Memory (OOM) errors, making a direct side-by-side measurement of wall-clock time or power consumption infeasible. We quantify the computational overhead introduced by AccLoRT's accumulation steps. The following table below shows the total wall-clock time under different ranks and accumulation intervals (frequency $T$) on a standard single-GPU (A100 40G) setup:
>
> frequency / rank | 50 |	100 | 150 | 200|
> |-|-|-|-|-|
> |1000|	2h13m|	2h17m|	2h21m|	2h17m |
> |2000|	2h6m|	2h10m|	2h14m|	2h9m |
> |3000|	2h2m|	2h6m|	2h10m|	2h6m |
> |4000|	2h2m|	2h6m|	2h10m|	2h6m |
> |5000|	2h2m|	2h6m|	2h10m|	2h5m |
>
> The results demonstrate that the overhead is minimal. Even with frequent accumulations (e.g., $T=1000$).
>
> We have added this discussions in the Appendix B.7.

---

> > ### Comment · Reviewer_UsMi · 2025-11-24
> > **Response to Authors**
> >
> > Thank you for the detailed explanation of my previous questions and concerns. I have already raised the score.

---

> > > ### Author Response · Authors · 2025-11-26
> > > **Response to Reviewer UsMi**
> > >
> > > We sincerely thank the reviewer for the positive feedback and the decision to raise the score.
> > >
> > > We are glad that our responses have addressed your concerns. We appreciate the time and effort you dedicated to reviewing our work, and your constructive comments have significantly helped improve the quality of our paper.

---

### Official Review · Reviewer_Kmhq · 2025-11-02

**Soundness:** 2
**Presentation:** 3
**Contribution:** 2
**Rating:** 4
**Confidence:** 3

**Summary:**

The paper introduces AccLoRT, a method for memory-efficient pre-training of LLMs with fully low-rank training. The core mechanism is a periodic accumulation and re-initialization strategy, which trains the adapters from scratch without initial pretrained weights, merges into a fixed high rank matrix, and continues with reinitialized adapters. The paper provides a theoretical basis which shows that the accumulated low-rank training is equivalent to full-rank training after a finite number of accumulation steps.

Experiments on pretraining of Llama models and finetuning of RoBERTa-base models validate the method.

**Strengths:**

- Despite a simple strategy, the method achieves good results across pretraining setup.
- Some discussion and experiments on the initialization of low rank matrices.
- Extensive experiments including those that help understand training progression.

**Weaknesses:**

- For large model pretraining, AccLoRT did not achieve significant gain over GaLore. Although the memory usage is lower at early stage, AccLoRT approaches the same memory usage level as GaLore if using the same rank. How does the extra memory at early stage benefit training?
- Would the plateau towards the end of each training cycle slows down the training in general and complicate the choice of ranks and iteration steps? Could the authors provide perplexity progression throughout training and compare with GaLore.
- The method is not efficient in fine-tuning. The performance is sometimes not as good as LoRA despite a larger adapter size (8 x num iterations). Also, the accumulation frequency varies across settings. What is the principal for choosing the rank and accumulation frequency? How sensitive the results are for these choices of hyper-parameters?

**Questions:**

- In table 1, for LoRA, should the total be + 4r(n + m) for both AccLoRT and LoRA?
- L450: "In the 1B parameter setting, while GaLore achieves the best perplexity of 15.64, AccLoRT maintains competitive performance at 16.61.", but in the table there's no 16.61 but 15.49. Is it a typo?
- Since ReLoRA is a very close approach, could you add more discussion for the difference in methodology that leads to the improvement. It seems that ReLoRA is only mentioned as requiring a full parameter warm up pretraining.

---

> ### Author Response · Authors · 2025-11-22
> **Response to Reviewer Kmhq (1/3)**
>
> Dear Reviewer Kmhq, we appreciate your insightful comments. In the following response, we address each of your concerns point-by-point and have updated the paper to reflect these changes. We welcome any further questions you may have.
>
> ---
> > **(Weakness 1)** *For large model pretraining, AccLoRT did not achieve significant gain over GaLore. Although the memory usage is lower at early stage, AccLoRT approaches the same memory usage level as GaLore if using the same rank. How does the extra memory at early stage benefit training?*
>
> **Re**: Yes, AccLoRT requires lower memory usage at the early stage. Previously, the extra memory saving has not contributed to the training. Indeed, it can be benefited to the training if we increase the rank of AccLoRT in the early stage rather than keeping a constant rank increase of r, and then decrease the smaller one to approach the memory of GaLoRe. In detailed, according to Table 1, the rank for AccLoRT in different accumulation stage $d$ can be given as
> $$
> r_d = \max\left( r + \frac{mn}{3(m+n)} - \frac{\sum_{i=0}^{d-1} r_i }{ 3 }, r\right),
> $$
> where $r$ denotes the pregiven rank. Therefore, to maintain consistent memory usage, we can let the rank of AccLoRT to be relatively large in the early stage, which can further improve the performance of AccLoRT. In the following, we present the performance of AccLoRT with the above adaptively rank strategy for pretraining LLaMA 60M model.
> We can observe that with adaptive rank in the early stage, the AccLoRT can be further improved.
>
> Method | 60M |
> |-|-|
> |AccLoRT | 33.77	|
> |AccLoRT (w. adaptive)| **33.34**	|
>
>
> ---
> > **(Weakness 2)** *Would the plateau towards the end of each training cycle slows down the training in general and complicate the choice of ranks and iteration steps? Could the authors provide perplexity progression throughout training and compare with GaLore.*
>
> **Re**: **Plateau effect does not affect the model**. We thank the reviewer for this question, we found that in practical experiment the plateau effect does not affect the model.
> We present the training curve in Figures 10 and 11 in the appendix, which shows that the 'plateau' effect from our toy example (r=5 in Figure 6) does not happen at these practical training (r=100, r=128).
> This is not surprising as we set a very small rank (r=5 in Figure 6) the model would take only a small amount of training steps to converge to its maximum capacity, that is, the LLM would gather enough information from the data to stabilise in this region. Please see Figures 10 and 11 for more details.
>
> **Loss curves and Eval. PPL progression on different rank and frequency**. In Figures 10 and 11, the Training Loss plots (left side) show a smooth, continuous decrease for all frequencies, proving that training efficiency is not slowed down the convergence rate. The Eval. PPL plots (right side) show that while there are minor variations (a slight divergence) between frequencies in the early stage (around 2000-4000 steps), all curves quickly re-converge and are very tightly clustered, reaching the same final performance. This demonstrate that AccLoRT is robust to different frequency setting. Please see Figures 10 and 11 for more details.
>
> **PPL progression compared with Galore**. Regarding to the comparison with Galore in the training process, we have added Figure 8 in Appendix to the revised paper, which directly compares the training progression of AccLoRT and GaLore on LLaMA 60M. The training loss plot (left) shows that AccLoRT's convergence curve is nearly identical to GaLore's, confirming no "plateaus" issue in each training cycle. Furthermore, the Eval. PPL plot (right) shows that AccLoRT consistently achieves a lower (better) PPL than Galore after the 4000-step, leading to superior final performance. Please see Figure 8 for more details.

---

> ### Author Response · Authors · 2025-11-22
> **Response to Reviewer Kmhq (2/3)**
>
> > **(Weakness 3)** *The method is not efficient in fine-tuning. The performance is sometimes not as good as LoRA despite a larger adapter size (8 x num iterations). Also, the accumulation frequency varies across settings. What is the principal for choosing the rank and accumulation frequency? How sensitive the results are for these choices of hyper-parameters?*
>
>
>  **Re**: We thank the reviewer for carefully examining the fine-tuning results and the hyperparameter choices for AccLoRT.
>
> **Fine-tuning vs. pre-training**: On GLUE with RoBERTa-base, AccLoRT attains the best average score, but the gains over LoRA are indeed modest (Table 4). This behavior is consistent with prior observations that downstream fine-tuning is often intrinsically low-rank, so standard LoRA already operates close to the optimum in many NLU benchmarks. In line with this, our main goal is to make pre-training memory-efficient, and we view AccLoRT in fine-tuning as an optional mechanism.
> We will clarify this scope and recommendation more explicitly in the revised paper.
>
> **Additional fine-tuning on a higher-rank task (GSM8K)**. To further evaluate AccLoRT in a setting where higher-rank adapters tend to be more beneficial, we have conducted additional experiments on GSM8K (rank=8, batch size=4), a more challenging mathematical reasoning benchmark that is empirically more sensitive to the adapter rank. We compare LoRA and AccLoRT under the same memory budget, varying both rank and accumulation frequency. The new results show that AccLoRT consistently outperforms LoRA once we allow a slightly higher effective rank via a small number of accumulations.
>
>
> | Method | Acc |
> |-|-|
> |LoRA | 47.84 |
> |AccLoRT (T=600) | 54.44|
>
> **Principle for choosing rank and accumulation frequency**. Our theoretical analysis provides a principled guideline. In pretraining, for an AccLoRT layer with rank $r$ and frequency $T$, the effective rank of the accumulated weight grows as $\text{rank}(W_t) = \lfloor t / T \rfloor r$, and, by Proposition 4.1, the rank of the sum of independent low-rank terms equals $\min(n,m,rd)$, where $d$ is the number of accumulations. Thus, given a total number of training steps, we first choose $r$ under a memory/throughput budget and then select frequency $T$ so that $T$ steps $\lceil t / T \rceil r \geq \min(m,n)$ by the end of training.
> In our pretraining experiments, a common practice was to set the rank to approximately 10\% of the average matrix size, and the accumulation frequency based on the total training steps $t_{\text{total}}$. Specifically, we define it as:
> $$
> T = \frac{t_{\text{total}}}{n/r} \times C,
> $$
> where $t_{\text{total}}$ is the number of total training steps, $n$ is the average matrix size, $r$ is the low-rank value, and $C \in [0.6, 1]$ is a scaling constant depending on the target convergence behavior.
>
> In the newly added Figures 10 and 11, we have investigated the stability of this strategy, which shows that giving $r=100$, the frequency $T$ should be $[1000, 2000]$ and giving $r=128$, the frequency $T$ should be $[1500, 2500]$. In Figure 10 and Figure 11, we show that we can obtain very stable performance when using the criterio by selecting the frequency in this interval. This setting has also shown to be **effective across various models** in our experiments.
>
> **Sensitivity of the method to these hyperparameters**. Empirically, we find that AccLoRT is robust within the frontier region: moderate changes in $r$ or $T$ lead to very small differences in perplexity, and proper ranks (e.g., 100 or 200) make performance even less sensitive to the exact frequency (Figure 7, Figure 10 and Figure 11). In contrast, when $r$ is too small or $T$ is too large such that the final effective rank $\lfloor \text{steps}/T \rfloor r$ falls below $\min(n,m)$, perplexity degrades as expected.
>
> Fine-tuning insensitivity to update frequency. On GSM8k with Mistral‑7B (rank = 8, batch size=4), sweeping the AccLoRT update frequency across a wide range (100–800) produces nearly identical accuracy. This flat response indicates that once a reasonable rank is fixed, the refresh frequency becomes a second-order hyperparameter in fine-tuning, i.e., we can simply pick any mid-range value (e.g., 400–700) without per-task tuning.
>
> | Frequency (r=8) | GSM8K Exact Match Acc. |
> |-----------|-------------------|
> | 100 | 53.22 |
> | 200 | 53.68 |
> | 250 | 52.99 |
> | 300 | 51.48 |
> | 400 | 52.08 |
> | 500 | 53.68 |
> | 600 | 54.44 |
> | 700 | 54.28  |
> | 800 | 53.83  |
>
> We have presented these discussions and results in Appendix B.2 and B.6. Please refer to the revised paper for more details.

---

> ### Author Response · Authors · 2025-11-22
> **Response to Reviewer Kmhq (3/3)**
>
> > **(Question 1)** *In table 1, for LoRA, should the total be + 4r(n + m) for both AccLoRT and LoRA?*
>
> **Re**: Thank you very much for carefully checking the parameter counts in Table 1, and we apologize for the confusion caused by the previous version. In the original table, we aggregated frozen and trainable weights into a single “Weight parameters” term, which made this convention unclear and could be misleading. In the revised paper, we have updated Table 1 to (i) explicitly separate "Frozen parameters" and "Trainable parameters", and (ii) clarify in the caption that the total footprint is computed as "frozen parameters + trainable parameters + optimizer states + gradients".
>
> ---
>
> > **(Question 2)**
> *L450: "In the 1B parameter setting, while GaLore achieves the best perplexity of 15.64, AccLoRT maintains competitive performance at 16.61.", but in the table there's no 16.61 but 15.49. Is it a typo?*
>
> **Re**: Thanks for pointing out the typo in our paper, it should be 15.49, we have revised it in the paper.
>
> ---
>
> > **(Question 3)** *Since ReLoRA is a very close approach, could you add more discussion for the difference in methodology that leads to the improvement. It seems that ReLoRA is only mentioned as requiring a full parameter warm up pretraining.*
>
> **Re**: We appreciate the reviewer’s observation that ReLoRA is conceptually related. There are several core methodological and mathematical differences that we believe are crucial for both our improvements and the novelty of AccLoRT. We summarize in the following and also presented in the revised paper (see page 7 for more details).
>
> **(1) Fully low-Rank training and subspace filling dynamics**. AccLoRT’s sequential low-rank training strategy fundamentally differs from ReLoRA, yielding distinct advantages in 1) memory efficiency and 2) training dynamics. Unlike ReLoRA, which relies on a full-parameter warmup, AccLoRT is a fully low-rank paradigm. Instead of refining weights in random directions after a heavy warmup, AccLoRT explicitly fills the parameter space by sequentially introducing low-rank subspaces. This allows the model to prioritize and optimize the most informative subspaces first at each accumulation step. This learning dynamic not only eliminates the high memory costs associated with full-rank warmups but also facilitates faster and more stable convergence (as evidenced in Section 5.1 and Figure 5).
>
> **(2) Fundamental difference in initialization**. ReLoRA relies on LLaMA's original initialization for the warmup stage and subsequently restarts adapters with standard Kaiming (for $A$) and zeros (for $B$). By contrast, AccLoRT eliminates the need for full-rank warmup. At $t=0$, we sample a Gaussian full-rank weight matrix (matching LLaMA’s distribution), perform a truncated QR decomposition, and set the initial $B=R$ and $A=Q$. For subsequent accumulations, we generate a new random matrix and orthogonalize it via truncated-QR to define a fresh, orthonormal subspace for $B$ (with $A$ initialized to zero). This initialization strategy is critical as it allows AccLoRT to strictly maintain low-rank constraints while maintains significantly faster loss convergence compared to standard Kaiming/Normal initializations as shown in Figure 3.
>
> **(3) Theoretical support for hyperparameter selection**. Unlike ReLoRA, which relies primarily on empirical observations, AccLoRT establishes a rigorous theoretical framework that guides practical usage. AccLoRT provides a theoretical analysis of rank evolution: we derive upper/lower bounds for the rank of sums of low-rank updates and prove that the effective rank grows linearly with the number of accumulations. This allows us to introduce a full-rank Pareto frontier in the (rank, frequency) plane. This theoretical frontier provides a principled rule for selecting hyperparameters. More details and ablations are presented in Appendix B.2 and Figures 10 and 11.
>
>
> **(4) Practical simplicity and stability**. While ReLoRA relies on a complex combination of heuristics to prevent divergence, AccLoRT offers a significantly more robust training pipeline. ReLoRA's strict recipe requires several non-trivial components: 1) a full-parameter warmup, 2) jagged cosine learning-rate schedules, and 3) magnitude pruning-based optimizer resets. Its own ablations show that omitting parts of this recipe often leads to degraded training. In contrast, AccLoRT eliminates these dependencies: we operate without full-rank training, jagged schedules, or manual optimizer pruning. Instead, we simply select rank and accumulation frequency based on our rank-evolution theory. This simplicity allows AccLoRT to achieve consistent convergence without extensive hyperparameter engineering, making it a more reliable and reproducible pretraining paradigm.
>
> Consequently, these methodological improvements shift the paradigm from heuristic-based adaptation to principled low-rank learning, directly accelerating convergence while minimizing memory costs.

---

### Author Response · Authors · 2025-12-03
**Response Summary**

We would like to thank AC for handling this manuscript and the reviewers for their thoughtful and constructive feedback.

The **main contribution** of this paper is to propose a fully low-rank pretraining paradigm, supported by a theoretical analysis of rank evolution. In line with this contribution, the reviewers **have highlighted several strengths** of the paper, including the characterization of rank evolution through our upper and lower bounds, the practical ease of implementation, and the strong empirical performance achieved under a significantly reduced memory footprint.

Two reviewers have already confirmed that our rebuttal resolved their concerns and raised their scores to **8** and **6**. Due to this year’s special circumstances, the remaining two reviewers were not able to respond further. Below, we summarize their remaining concerns and how our rebuttal addressed them.

**Reviewer Kmhq (no post rebuttal reply)**

1. **Request for evidence that early-stage memory savings translate into training benefits**: We have added experiments with an adaptive-rank strategy showing that the extra early-stage memory can be used to increase the effective rank, improving perplexity. (see newly added experiments and analysis in Appendix B.5 and Table 5).

2. **Concerns about potential plateau within each accumulation cycle**: We have included full training curves and PPL progression. These show no plateau in practical settings. AccLoRT’s convergence matches GaLore’s and achieves better PPL (see newly added experiments and analysis in  Appendix B.1 and Figure 9).

3. **Concern about fine-tuning effectiveness relative to LoRA**: We clarified that NLU benchmarks are intrinsically low-rank and LoRA is already near-optimal. We have added a GSM8K showing that, under the same memory budget, AccLoRT consistently outperforms LoRA as a higher effective rank is beneficial (see newly added experiments and analysis in Appendix B.6 and Table 6).

4. **Need for a clearer principle for selecting rank and accumulation frequency**: We have expanded the theory of rank evolution and provided a principled guideline for selecting frequency in Appendix B.2. Experiments on pretraining (see newly added Figures 10 and 11) and finetuning (see newly added Table 7) show that AccLoRT is robust across a wide frequency range.

5. **Typos in reported numbers**: We corrected the typo of 1B perplexity and revised Table 1 by separating frozen parameters, trainable parameters, optimizer states, and gradients for clarity.

**Reviewer Et4y (no post rebuttal reply)**

1. **Misunderstanding about the initialization strategy**: We have clarified that the model is not initialized to zero in the revised paper.

2. **Request for missing assumptions in Theorem 3.3**: We have added an explicit claim on the initialization of $A$ and $B$, explained why matrices drawn from continuous distributions are almost surely full-rank, and clarified why hidden states are linearly independent with high probability.

3. **Request for clearer methodological differences from ReLoRA**: The reviewer requested a more explicit explanation of how AccLoRT compares to ReLoRA. We have addressed this by detailing that:
	- AccLoRT removes the full-rank warm-up entirely,
	- introduces a sequential subspace learning paradigm,
	- provides a full theoretical framework for rank evolution.

    These additions more clearly distinguish the two methods. We have added detailed discussions in Section 4.2.

4. **Request to explain why AccLoRT can outperform ReLoRA**: We have clearly explained the difference in learning dynamics: ReLoRA refines a full-rank warmup model, while AccLoRT gradually fills the parameter space through low-rank subspaces, which leads to faster and more stable convergence in practice.

5. **Need to clarify the purpose of the if–else logic in Algorithm 1**: We clarified that this mechanism enables early-stage memory savings, which in turn allows an adaptive rank strategy for using larger ranks early without exceeding memory limits.

**Conclusion**

For the remaining two reviewers who were unable to reply, their concerns mainly focused on the need for stronger empirical evidence regarding training dynamics, hyperparameter choices, and fine-tuning behavior, and the clarity of the initialization strategy and theoretical assumptions, and the distinction between AccLoRT and related methods such as ReLoRA. Our rebuttal provided comprehensive responses supported by newly added deeper analysis and experiments, including full training curve results, adaptive rank experiments, frequency selection studies, and fine-tuning on GSM8K. We also strengthened the clarity of the paper by explicitly distinguishing AccLoRT from ReLoRA, completing the theoretical assumptions behind our rank-evolution analysis, and refining the explanation of our initialization.

Finally, we thank the AC for their time and careful evaluation of our submission.

Sincerely,

Authors of Submission 9174

---

### Meta-Review · Area_Chair_rJfo · 2026-01-04

**Summary:**

The paper proposes AccLoRT, a memory-efficient pre-training method that sequentially trains and accumulates multiple low-rank matrices. This eliminates the need for full-parameter warm-up. The motivation and method design are supported by theoretical analysis. However, one key outstanding concern is the absence of SOTA baselines. All main baselines are published in or before 2024, and more recent methods, such as Fira mentioned by reviewers, are not included. The result reported in the paper of Fira shows Fira significantly outperforms AccLoRT. Moreover, two reviewers are concerned that the algorithmic novelty is incremental. Given these unresolved issues, the paper falls into the borderline rejection.

**Reviewer Concerns:**

The rebuttal addresses concerns on
1) early-stage memory benefits by providing adaptive rank experiments showing memory can improve perplexity; 2) training plateaus by showing no plateau in practice;
3) difference from ReLoRA with detailed methodological contrasts; and
4) clarity on initialization and assumption in Theorem 3.3.

Two concerns are partially addressed.
1) For hyperparameter selection principles, authors add theoretical guideline with empirical validation. But it may still be heuristic in practice.
2) For the fine-tuning effectiveness compared to LoRA, although the authors showed improvements on GSM8K, they conceded that for many NLU tasks, LoRA is already near-optimal.

Outstanding concerns including
1) Comparison with Fira: only discussion added without experimental comparison;
2) The algorithmic novelty is incremental compared to ReLoRA, though authors provided explanation;
3) Full 7B comparisons with baselines such as Fira are not reported, possibly due to incomplete results in the rebuttal.

**Reviewer Scores:**

Reviewer UsMi raised score to 8 from 6.

Reviewer 13sN raised score to borderline accept, but noted comparison with FIRA at 7B scale is needed. The final score may be 4 or 6.

Reviewer Kmhq may raise to borderline accept, given most concerns were experimentally addressed. However, some skepticism about fine-tuning effectiveness and methodological novelty may remain.

Reviewer Et4y might increase their score to 6 given the clarifications on initialization, assumptions, and differences from ReLoRA, but may remain at 4 if they continue to view the contribution as incremental.

---

### Decision · Program_Chairs · 2026-01-26

Reject